# Image3C, a multimodal image-based and label-independent integrative method for single-cell analysis

Alice Accorsi[1,2‡], Andrew C Box[1‡], Robert Peuß[1,3‡], Christopher Wood[1], Alejandro Sánchez Alvarado[1,2†]*, Nicolas Rohner[1,4†]

[1]Stowers Institute for Medical Research, Kansas City, United States; [2]Howard Hughes Medical Institute, Stowers Institute for Medical Research, Kansas City, United States; [3]Institute for Evolution and Biodiversity, University of Münster, Münster, Germany; [4]Department of Molecular and Integrative Physiology, KU Medical Center, Kansas City, United States

**Abstract** Image-based cell classification has become a common tool to identify phenotypic changes in cell populations. However, this methodology is limited to organisms possessing well-characterized species-specific reagents (*e.g.*, antibodies) that allow cell identification, clustering, and convolutional neural network (CNN) training. In the absence of such reagents, the power of image-based classification has remained mostly off-limits to many research organisms. We have developed an image-based classification methodology we named Image3C (Image-Cytometry Cell Classification) that does not require species-specific reagents nor pre-existing knowledge about the sample. Image3C combines image-based flow cytometry with an unbiased, high-throughput cell clustering pipeline and CNN integration. Image3C exploits intrinsic cellular features and non-species-specific dyes to perform *de novo* cell composition analysis and detect changes between different conditions. Therefore, Image3C expands the use of image-based analyses of cell population composition to research organisms in which detailed cellular phenotypes are unknown or for which species-specific reagents are not available.

**\*For correspondence:**
asa@stowers.org

[†]These authors contributed equally to this work
[‡]These authors also contributed equally to this work

**Competing interests:** The authors declare that no competing interests exist.

## Introduction

Single-cell analysis has proven crucial to our understanding of fundamental biological processes such as development, homeostasis, regeneration, aging, and disease (*Goolam et al., 2016*; *Kimmel et al., 2019*; *Pepe-Mooney et al., 2019*; *Philippeos et al., 2018*; *Tirosh et al., 2016*). High-throughput analyses of these and other biological processes at single-cell resolution require technologies capable of describing individual cells and subsequently clustering them based on similarities of features like morphology, cell surface protein expression, or transcriptome profile. Recent advances in image-based cell profiling and single-cell RNA sequencing (scRNA-seq) allow quantification of differences between cell populations and comparisons of cell type composition between samples (*Caicedo et al., 2017*). Single-cell studies that use traditional research organisms (*e.g.*, mouse, rat, or fruit fly) benefit from the availability of genomic platforms and established antibody libraries. However, the same cannot be said for a growing number of important, yet understudied research organisms lacking such reagents and whose biological interrogation would benefit immensely from single-cell analyses. In these cases, classical histochemical methods are often used to identify and characterize specific cells. Yet, the successful identification and enumeration of biologically meaningful cell types in such studies can be harmed by both the limited number and variety of cellular attributes (few features or low dynamic range) available for determination of cell types and by observer bias when using traditional, hand-counting approaches (*e.g.*, hemocytometer and Giemsa stain)

**eLife digest** Cells are the building blocks of all living organisms. They come in many types, each with a different role. Understanding the composition of cells, *i.e.*, how many cells and which types of cells are present inside an organ can indicate what that organ does. It can also reveal how that organ changes under different conditions, like during an infection or treatment.

The most powerful methods for studying cells work well for species researchers already know a lot about, such as mice, zebrafish or humans, but not for less studied animals. To change this Accorsi, Box, Peuß et al. created a new tool called Image3C to be used for studying the composition of cells in less researched organisms.

Instead of using reagents that only work for specific species, the tool uses molecules that work across many species, like dyes that stain the cell nucleus. A cell-sorting machine, known as a flow cytometer, connected to a microscope then takes pictures of hundreds of stained cells each second and Image3C groups them based on their appearance, without the need for any prior knowledge about the cell types. Accorsi et al. then tested Image3C on immune system cells of zebrafish, a well-studied animal, and apple snails, an under-studied animal. For both species, the tool was able to sort cells into groups representing different parts of the immune system.

Image3C speeds up the grouping process and reduces the need for user intervention and time. This lowers the risk of bias compared to manual counting of cells. It can sort cells even when the types of cells in an organism are unknown and even when specialized reagents for an organism do not exist. This means that it could characterise the cell make-up of new tissues coming from organisms never studied before. Access to this uncharted world of cells stands to reveal previously inaccessible clues about how organs behave and evolve and allow researchers to investigate the impact of environmental changes on these cells.

(*van der Meer et al., 2004*). These shortcomings, together with the lack of extensive knowledge on cell-specific phenotypes available for training or for *a priori* assumptions, usually result in the under-estimation of the complexity of cellular composition or interactions among cell types within tissues.

Automated classification of cells using convolutional neural networks (CNNs, machine learning [ML] method specialized in image recognition and classification) has become a promising approach for accurate high-throughput cell analysis that is free from observer bias (*Blasi et al., 2016*; *Eulenberg et al., 2017*; *Kobayashi et al., 2017*; *Lei et al., 2018*; *Nassar et al., 2019*; *Suzuki et al., 2019*). To date, CNN-based automated clustering and classification techniques require pre-existing knowledge about the organism or cell type of interest (*e.g.*, cell-specific morphological traits within an image set) or the availability of cell-specific reagents (*e.g.*, antibodies), or genomic sequence (*e.g.*, single-cell sequencing) (*Table 1* shows an overview of the existing methods) (*Baron et al., 2019*; *Blasi et al., 2016*; *Cheng et al., 2021*; *Eulenberg et al., 2017*; *Hennig et al., 2017*; *Kobayashi et al., 2017*; *Lei et al., 2018*; *Nassar et al., 2019*). This means that to make effective use of artificial intelligence (AI) approaches for single-cell analysis, one must have information available to train the algorithm or for ML models, which often arises in the form of information gleaned from the use of reagents like antibodies. Research areas that rely on inter-species comparisons or studies on emerging research organisms would benefit from single-cell-based analyses that do not require pre-existing knowledge of cell types (*i.e.*, which is required for training a CNN for example) and/or availability of antibodies or molecular databases. For example, within the interdisciplinary field of eco-immunology, a growing number of researchers are investigating immune system adaptation to different environments by studying immune cell compositions in diverse animals (*Maizels and Nussey, 2013*). Given the influences of immune cell composition on the immune system response of an organism (*Kaczorowski et al., 2017*), applying modern single-cell analysis in eco-immunological research would substantially increase our knowledge about the plasticity and conservation of immune responses in a variety of different animals and conditions (*Peuß et al., 2020*).

To make sophisticated cellular composition analysis available to any research organism without the need for either pre-existing knowledge about the cell populations or species-specific reagents, we developed Image-Cytometry Cell Classification (Image3C). This method analyzes, visualizes, and quantifies, in a high-throughput and unbiased way, the composition of cell populations by using cell

**Table 1.** Extensive overview on label-free cell clustering tools including a comparison of their main features.

| Tool name | Was tested on multiple cell types? | Requires *a priori* knowledge of the sample and/or species-specific reagents for the clustering? | Uses commercially available hardware? | Uses free or open source softwares? |
|---|---|---|---|---|
| Image3C (present work) | YES (Zebrafish whole kidney marrow; Apple snail hemolymph) | NO Does not require previous knowledge or species-specific reagents to cluster cell images and train the neural network. If available, they can be used/integrated. | YES (ImageStream) | YES (IDEAS; Vortex; R; Python) NO (FCS Express Plus) |
| CellProfiler [*] | YES (Fixed Jurkat cells; Live Jurkat cells; Fission yeast; Human white blood cells) | YES Requires annotated datasets to train the machine learning algorithms, either by staining the samples with known markers, or by manually clustering the cells. | YES (ImageStream; Microscope) | YES (IDEAS; CellProfiler) NO (MATLAB) |
| CellProfiler Analyst [†] | NO (Jurkat cells) | YES Requires the use of fluorescent markers to annotate the cells and use them as the ground truth to train the machine learning algorithms. | YES (ImageStream) | YES (IDEAS; CellProfiler/ Phyton) |
| Label-free reflectance microscopy [‡] | NO (Fixed HeLa cells) | YES Requires immunofluorescence images with known markers to use as ground truth and for training multiple deep learning models. | NO (Custom-built multimodal light-emitting diode (LED) array reflectance microscope) | YES (Deep neural networks) |
| Optofluidic time-stretch microscopy [§] | NO (Human breast adenocarcinoma cell line, MCF-7) | YES Does not provide cell clustering and single-cell resolution analysis. The changes are analyzed overall in the sample without assigning it to a cell type. | NO (Optofluidic time-stretch microscope and microfluidic devices) | NO (MATLAB) |
| Raman scattering [¶] | YES (Microalgal cells; Circulating tumor cells in human blood) | YES Requires homogenous cell cultures. After different treatments, these samples are used to create databases for training deep learning models. | NO (High-speed multicolor stimulated Raman scattering (SRS) microscope and microfluidic platform) | YES (Deep learning, neural network structure, VGG-16) |
| Fluorescence lifetime imaging [**] | NO (Human white blood cells) | YES Depends on other techniques to identify cell types to compare their autofluorescence signals and fluorescence decay for further analysis. | YES (Fluorescence microscopy; FLIM; Flowcytometer) | YES (Python) |

[*] *Blasi et al., 2016*; *Nassar et al., 2019*.

[†] *Hennig et al., 2017*.

[‡] *Cheng et al., 2021*.

[§] *Kobayashi et al., 2017*.

[¶] *Suzuki et al., 2019*.

[**] *Yakimov et al., 2019*.

morphological traits and non-species-specific fluorescent probes (*e.g.,* nuclear staining or dyes for metabolic states that function well in a variety of organisms) (*Figure 1*, *Table 1*). By taking advantage of cell morphology and/or fluorescent dyes related to function or metabolic state, Image3C can analyze single cell suspensions derived from any experimental design, *de novo* cluster cells present in the sample of interest and compare their abundance between each other or among different assays. Once the *de novo* clustering based on cell-intrinsic features is obtained, Image3C employs a CNN that uses these clusters as training sets, avoiding in this way user bias or manual classification (*Figure 1*, *Table 1*). This produces a CNN-based cell classifier 'machine' used to quantify subsequently acquired image-based flow cytometry data and compare cellular composition of samples across multiple experiments in a high-throughput manner without the need for repeating time-consuming steps for *de novo* clustering. The combination of the clustered cell images, the outputs of their functional assays, and the published literature about closely related organisms might allow the identification and description of cell types of interest. In comparison to existing label-free cell clustering methods, Image3C does not require initial antibody staining (*Cheng et al., 2021*; *Hennig et al., 2017*; *Lippeveld et al., 2020*; *Nassar et al., 2019*), pre-existing knowledge of specific cell morphology (*Suzuki et al., 2019*; *Yakimov et al., 2019*), and is not limited to a specific

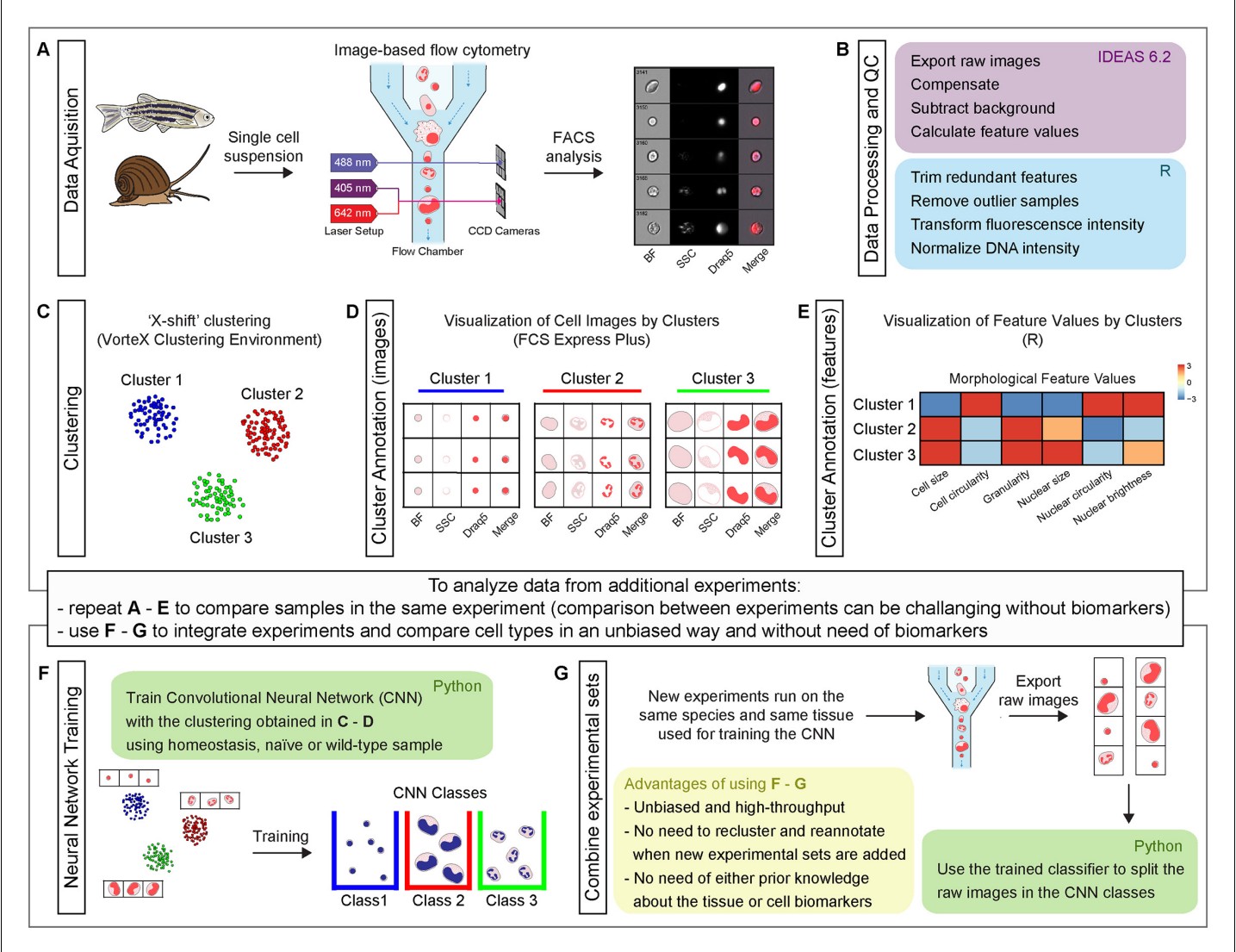

**Figure 1.** Schematic representation of Image3C workflow, a method for cell clustering based on morphological features. (**A**) A single cell suspension is prepared for image-based flow cytometric analyses. The cells can be labeled with any reagent working for the species of interest. The signal can highlight specific cell components (*e.g.*, nuclei), metabolic cell states, or specific cell functions. The samples are run on the ImageStream$^X$ Mark II, and 10,000 nucleated and focused events are saved for each sample as individual raw images. (**B**) IDEAS software is used to open the raw images, compensate for correcting fluorescent spillover, subtract background, and quantify values for intrinsic morphological and fluorescent features. R (or R studio) is used to calculate the correlation between features to allow to trim the features that are redundant with others. Samples that are outliers among replicates are also removed prior to the final normalization of the fluorescence intensities. (**C**) Images are clustered based on morphological and fluorescent feature values and visualized as a force-directed layout (FDL) graph where each dot represents one event. (**D**) R integration in FCS Express Plus software allows the visualization of cell images by clusters or specifically selected with a gate. This step allows to evaluate the morphological homogeneity of the clusters, determine if the number of clusters is appropriate, and explore the phenotype/function of the cells based on visualization of individual channels. (**E**) Spearman's correlation plot of feature values by clusters is one of the options available in Image3C for plotting integrated data. This heatmap shows feature similarities and differences between cells belonging to different clusters. (**F**) If new experiments are run and new data needs to be analyzed, two approaches can be taken. (1) If the goal is comparing samples belonging to the same experiment (*e.g.*, treatment vs. control), the steps described so far from (**A**) to (**E**) can be reapplied to the new dataset including a statistical analysis to compare cluster relative abundance. This approach will produce a new set of clusters that will need to be reannotated. Compare sets of clusters coming from multiple experiments and multiple rounds of analysis can be challenging without pre-existing knowledge of cell types, clearly different morphologies or biomarkers that would allow to establish a unique correlation between clusters coming from different FDL graphs. (2) If the goal is integrating experiments and comparing cell type abundance between them, the use of steps (**F**) and (**G**) is suggested. A CNN classifier is trained using the images obtained from homeostasis, naïve or wild-type (WT) cells, and already organized in clusters in an unbiased way through the first part of our method. This will generate a trained classifier with CNN classes based on FDL clusters. (**G**) This classifier is then used for deconvoluting data from new experimental sets and assigning each event to a CNN class with a given probability. This provides high-throughput and unbiased way to compare

*Figure 1 continued on next page*

*Figure 1 continued*

different experiment sets without the requirement for pre-existing knowledge about the tissue cell types, cell biomarkers, or the need to cross-annotate clusters increasing the probability to introduce errors. The entire pipeline chart and step-by-step technical information, such as software used, time required for processing, and exported file format, are reported in the interactive map *Figure 1—figure supplement 1* that automatically directs to the specific sections of the GitHub.

The online version of this article includes the following figure supplement(s) for figure 1:

**Figure supplement 1.** Interactive map of Image3C pipeline. Once the images are collected, this pipeline can be followed step-by-step.

cellular phenotype (*Blasi et al., 2016*) for *a priori* identification of certain cell types (*Table 1*). This makes Image3C extremely versatile and applicable to virtually any research organism and tissue from which dissociated single cells can be obtained. Parallelly to this *de novo* clustering approach, Image3C can take advantage of species-specific reagents and prior knowledge to be combined with transcriptomic dataset and provide a new and complimentary layer of information based on cell morphology and function. In sum, Image3C combines modern high-throughput data acquisition by image-based flow cytometry, advanced and unbiased clustering analysis, statistics to compare cellular compositions across different samples, and a CNN classifier component to easily determine changes in cell composition across multiple experiments.

## Results and discussion

### Image3C

Image3C is an imaging tool developed to study tissue composition at single-cell resolution in research organisms for which antibodies and pre-existing knowledge about cell types are not readily available (*Figure 1*, *Table 1*). Image3C allows for high-throughput and unbiased analysis in scenarios where manual counting and observer-based cell identification are currently the only options. Image3C includes all the components required for compensating captured images, quantifying multiple features for each event, clustering the events, visualizing and exploring the data, and training and using the CNN for analyzing subsequent samples and integrating multiple experiments (*Figure 1*, *Figure 1—figure supplement 1*).

Once a single cell suspension is prepared from the organism of interest, the cells are stained with a combination of dyes that are expected to function irrespectively of the species used and which have high affinity for specific cellular organelles such as nuclei or molecules associated with metabolic states such as reactive oxygen species (ROS). We validated reagents experimentally by determining that nuclear dyes stain intracellular material matching expected characteristics of nuclear DNA or by activation of cells with drugs to change their metabolic state. The labeled samples are then run on the ImageStream$^X$ Mark II (*Figure 1A*). ImageStream is a commercially available image-based flow cytometer, whose diffusion in laboratory settings is increasing and that provides highly reproducible images of cells that can be compared across days of acquisitions and experiments. For this approach, no microfluidic devices or custom-made and highly specialized microscopes are required (*Table 1*), and, if desired, the users can test the Image3C pipeline also on images acquired at standard microscopes, remembering to carefully control for batch effects.

Once images of individual events are collected for each channel of interest, feature values from both morphological and fluorescent data, such as cell size and nuclear size, are extracted from the cell images using IDEAS software (Amnis Millipore, free for download upon creation of Amnis user account) (*Figure 1B*; see *Supplementary files 1* and *2* for feature description). Correlation between features is calculated and redundant features are trimmed as well as samples that, among replicates, are outliers (*Figure 2—figure supplements 1* and *2*). This prevents clustering artifacts potentially caused by having multiple features providing the same information or including samples that are not representative (*Figure 1B*). During this step, while the number of features was usually reduced significantly, the correlation between replicates was always high and outliers were rarely observed (*Figure 2—figure supplements 1* and *2*). Finally, fluorescence intensity features are transformed to improve homogeneity of variance of distributions and, if used, DNA staining is normalized to remove intensity drift between samples and thus align the 2N and 4N DNA content histogram peaks (*Figure 1B*, *Figure 2—figure supplement 3*).

Exported feature quantifications are used for clustering the events. Dimensionality reduction and visualization of clusters is achieved by generating force-directed layout (FDL) graphs in the VorteX clustering environment (*Figure 1C*) (free to install) (*Samusik et al., 2016*). Cell images for events within each cluster can be visualized using FCS Express Plus together with custom R scripts (*Figure 1D*). These visualization tools and the cluster feature averages (*i.e.*, the mean value of each feature for each cluster) (*Figure 1E*) allow to explore the images of selected groups of events and the features that differ between cells belonging to separate clusters. If control and treatment samples are included, a statistical analysis using negative binomial regression to compare cell counts per cluster between samples is also available in the Image3C pipeline. This high-throughput and unbiased analysis provides a comprehensive understanding of a cell population composition at higher resolution than what is possible with traditional histological methods.

Once this pipeline is run on a first set of samples (*e.g.*, homeostatic state) and the cell clusters are defined for the tissue of interest, the images and the relative clustering IDs can be used to train a CNN classifier in an unbiased way (*Figure 1F*), including the ability to score frequency of 'new' cell types that do not match any of the clusters identified at homeostasis. Therefore, future experiments in the same tissue used for training the CNN classifier can be analyzed directly through the CNN (*Figure 1G*). This significantly reduces the number of steps and time required to process data collected subsequently. An even greater advantage is represented by the fact that, in the absence of CNN, every time new experimental sets are run it would be necessary to go again through the *de novo* clustering part of the pipeline (*Figure 1B–E*) and the new set of clusters would need to be cross-annotated to be compared with cell population composition observed in previous experiments. Manually matching clusters between different experimental sets might be a source of errors, mainly if the user is not familiar with the cell types present in the sample and if specific biomarkers or pre-existing knowledge about cell types and morphology are not available. The CNN splits all the cell images in the classes defined during the training step and allows to compare the abundance of cells with same morphology among different samples without the need to cross-annotate clusters (*Figure 1F, G*). The CNN inclusion in Image3C and the reproducibility of image acquisition through image-based flow cytometry allows use of the clusters defined from one experiment (*e.g.*, homeostatic state) to set up a classifier in an unbiased way for later use as a reference in analyzing the effects of experimental manipulations on these cell populations. We conclude from these results that Image3C can perform *de novo* high-throughput characterizations of population composition and define specific cell type changes between homeostatic and experimentally perturbed samples across multiple experiments.

## Image3C recapitulates cell composition of zebrafish whole kidney marrow (WKM) tissue

To test whether Image3C could identify homogeneous and biologically meaningful cell populations, we used the research organism *Danio rerio*. We obtained cells from adult female zebrafish WKM (location of the hematopoietic tissue) in homeostasis condition, stained them, and ran on the ImageStream[X] Mark II. We analyzed intrinsic morphological and fluorescent features, such as cell and nuclear size, shape, and darkfield signal (side scatter [SSC]). Feature values were extracted from each cell image and processed through our pipeline (see *Supplementary file 1* for feature description). Clustering by the final set of normalized and non-redundant morphological and fluorescent features produced distinct cell populations (*Figure 2A–C*, *Figure 2—figure supplements 1–3*).

Image3C can distinguish between the major classes of cells present in zebrafish WKM (*Figure 2*; *Supplementary files 3* and *4*) that were described using standard sorting flow cytometry and morphological staining approaches (*Traver et al., 2003*). It is noteworthy that Image3C can clearly identify dead cells and debris (*Figure 2A, B*) allowing to optimize experimental protocols in order to minimize cell death and run the subsequent analysis only on the intact, live cells. Image3C can identify cells with outstanding morphological features, such as neutrophils from other myelomonocytes (*Figure 2B, C*). Based on zebrafish neutrophil characteristics such as high granularity, high complexity, and low circularity of the nuclei (*Lugo-Villarino et al., 2010*), this type of granulocytes can be easily distinguished. Other types of myelomonocytes, such as monocytes and eosinophils, are here merged in the same cluster since in zebrafish they share many morphological characteristics (*Lugo-Villarino et al., 2010*). Similarly, using only intrinsic morphological features for the clustering, different lymphocytes (B and T-cells) and hematopoietic stem cells cannot be separated from each other,

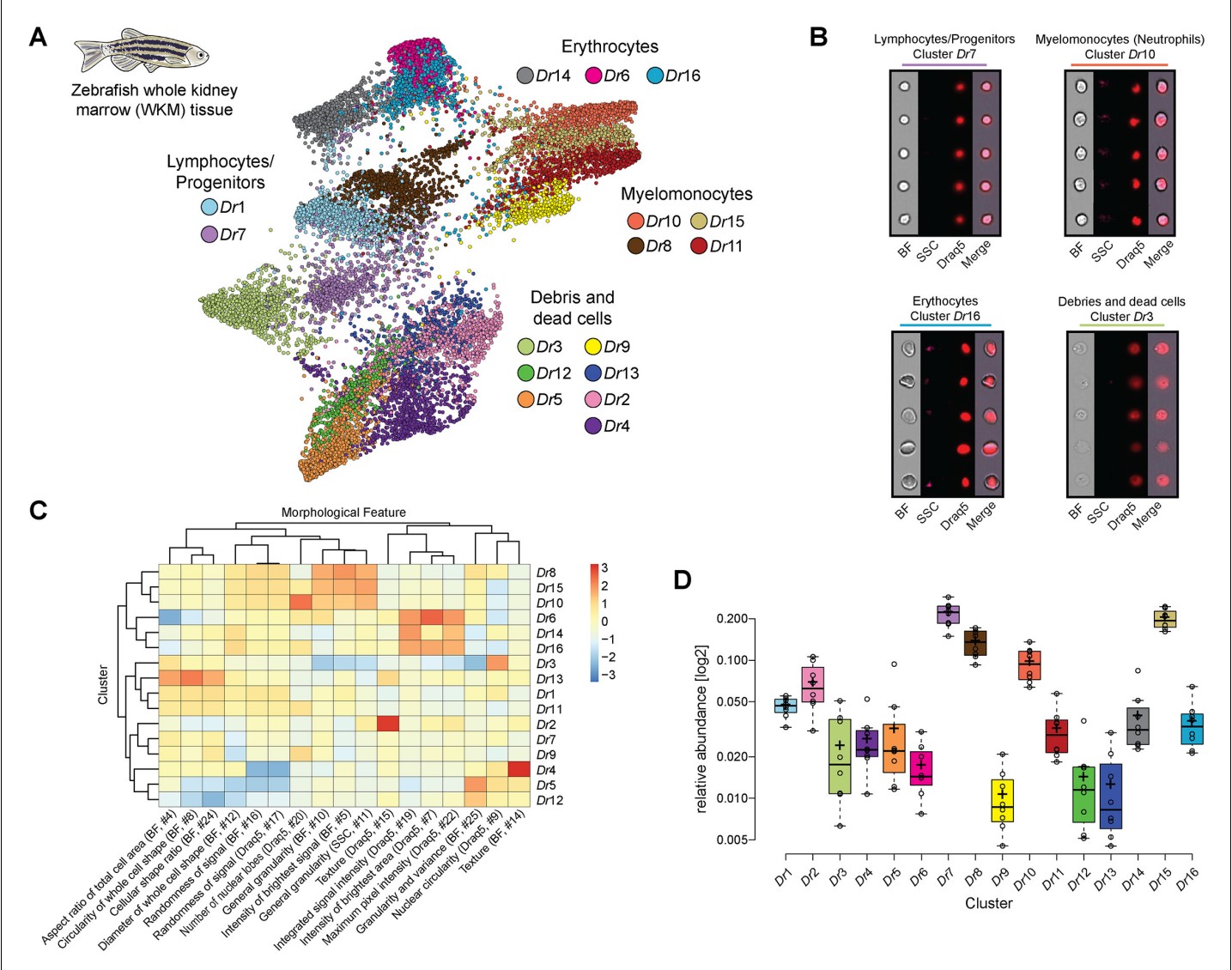

**Figure 2.** Analysis of cell composition of adult zebrafish whole kidney marrow (WKM). (**A**) WKM tissue obtained from zebrafish is prepared for image-based flow cytometric analyses and run on the ImageStream$^X$ Mark II (n = 8). Standard gating of focused and nucleated events and manual outgating of most erythrocytes was performed using IDEAS software. The selected images were processed through the pipeline described in *Figure 1* and clustered based only on intrinsic morphological and fluorescent feature values. Force-directed layout (FDL) graph visualizes 16 clusters, and each color represents a unique cell cluster. (**B**) Representative cell images belonging to each cluster are shown to evaluate the homogeneity of the cluster and determine morphology of the cells for cluster annotation. Representative cells for all the identified clusters are shown in *Supplementary file 4*. Merge represents the overlay of brightfield (BF) and Draq5 (nuclear staining) signal. (**C**) Spearman's correlation plot shows the average feature values of the images in each cluster to highlight morphological similarities and differences between events belonging to different clusters, such as cell size or cytoplasm granularity (*Supplementary file 1*). (**D**) Box plot of relative abundance of events within each cluster follows the same color code used in (**A**). The online version of this article includes the following figure supplement(s) for figure 2:

**Figure supplement 1.** Feature correlation for morphology assay on adult zebrafish whole kidney marrow (WKM).

**Figure supplement 2.** Sample correlation for morphology assay on adult zebrafish whole kidney marrow (WKM).

**Figure supplement 3.** DNA normalization for morphology assay on adult zebrafish whole kidney marrow (WKM).

but they can be clearly distinguished from the myelomonocytes (*Figure 2A, B*). Within the lymphocytes/progenitors fraction, we find two clusters (*Dr*1 and *Dr*7) that mainly differ in cell diameter (*Figure 2C*). Whether this morphological difference has a biological implication needs to be studied in future experiments.

Image3C also enables the quantification of cell populations (clusters or CNN classes)-relative abundance, an important tool for comparing population composition across different treatment groups under different environmental conditions (*Peuß et al., 2020*). Here, we compared our results with previously published data to validate our method. Although a direct comparison with results from classical approaches (*Traver et al., 2003*) is not possible since we gated out (removed analytically) mature erythrocytes before clustering (Materials and methods), the myelomonocyte to lymphocyte ratio (M/L ratio = 1.59) is similar to the one obtained with classic histological approaches (mean M/L ratio = 1.35) (*Figure 2D*; *Traver et al., 2003*).

## Image3C identifies professional phagocytes in zebrafish WKM tissue

Next, we sought to determine whether Image3C could be used to characterize and quantify biological processes by identifying a tissue of interest and then comparing cellular composition dynamics, functions, and physiological responses of specific cell types across a range of experimental conditions. Our goal was to detect statistically significant changes in cluster relative abundance between control and treated samples to gain a more detailed understanding of cell population dynamics and individual cell function.

As proof of concept, we performed a standard phagocytosis assay using WKM tissue from female adult zebrafish. The single cell suspension was incubated with CellTrace Violet labeled *Staphylococcus aureus* (CTV-*S. aureus*) and with dihydrorhodamine-123 (DHR), a ROS indicator that becomes fluorescent if oxidized to report oxidative bursting following phagocytosis. As controls, we inhibited phagocytosis through cytoskeletal impairment with cytochalasin B (CCB) incubation or through incubation with bacteria at lowered temperature by placing culture plates on ice. Events collected on the ImageStream$^X$ Mark II were analyzed with Image3C and clustered in 26 distinct clusters using quantifications of morphological and fluorescent features (see *Supplementary file 2* for feature description), including nuclear staining, phagocytized *S. aureus,* and DHR positivity (*Figure 3A*, *Figure 3—figure supplement 1*). Professional phagocytes are defined by their ability to take up *S. aureus* (CTV staining lies within the cell boundary) and induce a ROS response (bright DHR signal) (*Rabinovitch, 1995*). In zebrafish, professional phagocytes are mainly granulocytes and monocytic cells and can be discriminated from each other based on morphological differences, such as cell size, granularity, and nuclear shape (*Wittamer et al., 2011*). To compare samples incubated with CTV-*S. aureus* and the samples where phagocytosis is inhibited (CTV-*S. aureus* + CCB and CTV-*S. aureus* + Ice), we used the statistical analysis included in Image3C based on a negative binomial regression model (*Figure 3B, C*, *Figure 3—figure supplement 2*; *Supplementary files 5* and *6*). Statistical analyses reported clusters with differences in relative abundance between phagocytosis and phagocytosis-inhibited samples. Visualizing these clustered event images (*Supplementary file 7*) while considering the values and intensities of their morphological and fluorescent features (*Supplementary file 3*) allowed identification of 4 clusters of professional phagocytes: granulocytes within clusters *Dr*4_P, *Dr*12_P, and *Dr*13_P and monocytic cells in cluster *Dr*21_P (*Figure 3A, B*). The morphology of cells in cluster *Dr*12_P is characteristic of phagocytic neutrophils (*Figure 2B*, *Figure 3A*) that become adhesive and produce extracellular traps upon recognition of bacterial antigens (*Palić et al., 2007*). Overall, the relative abundance of professional phagocytes is 5–10% (*Figure 3C*), which is in line with previous studies that estimated the number of professional phagocytes in WKM tissue of adult zebrafish using classical morphological approaches (*Wittamer et al., 2011*). It is also noteworthy that in line with other studies (*Page et al., 2013*) we did not observe a cluster of lymphocytes (*e.g.*, B-cells) that actively phagocytize CTV-*S. aureus* bacteria (*Figure 2*; *Supplementary file 7*). Compared to the classical morphological approaches, Image3C allows to analyze thousands of events in a high-throughput and unbiased fashion, allowing the study of rare cell morphologies and increasing results confidence and reproducibility. These results show that Image3C can successfully analyze biological processes since we were able to recapitulate the presence, cell type, and frequency of professional phagocytes in adult zebrafish WKM organ.

A new aspect that Image3C highlighted is that CCB selectively affects cell viability based on cell identity, introducing artifacts and cell damage, actions not specific to inhibition of phagocytosis (*Figure 3B*). All mature erythrocyte-containing clusters had a significantly higher cell count in the CTV-*S. aureus* samples compared to the CTV-*S. aureus* + CCB ones (*Figure 3B*; *Supplementary files 3* and *5*). Cluster analysis revealed that erythrocytes were almost absent in samples incubated with CCB (*Supplementary file 3*), while there was a significant increase in the

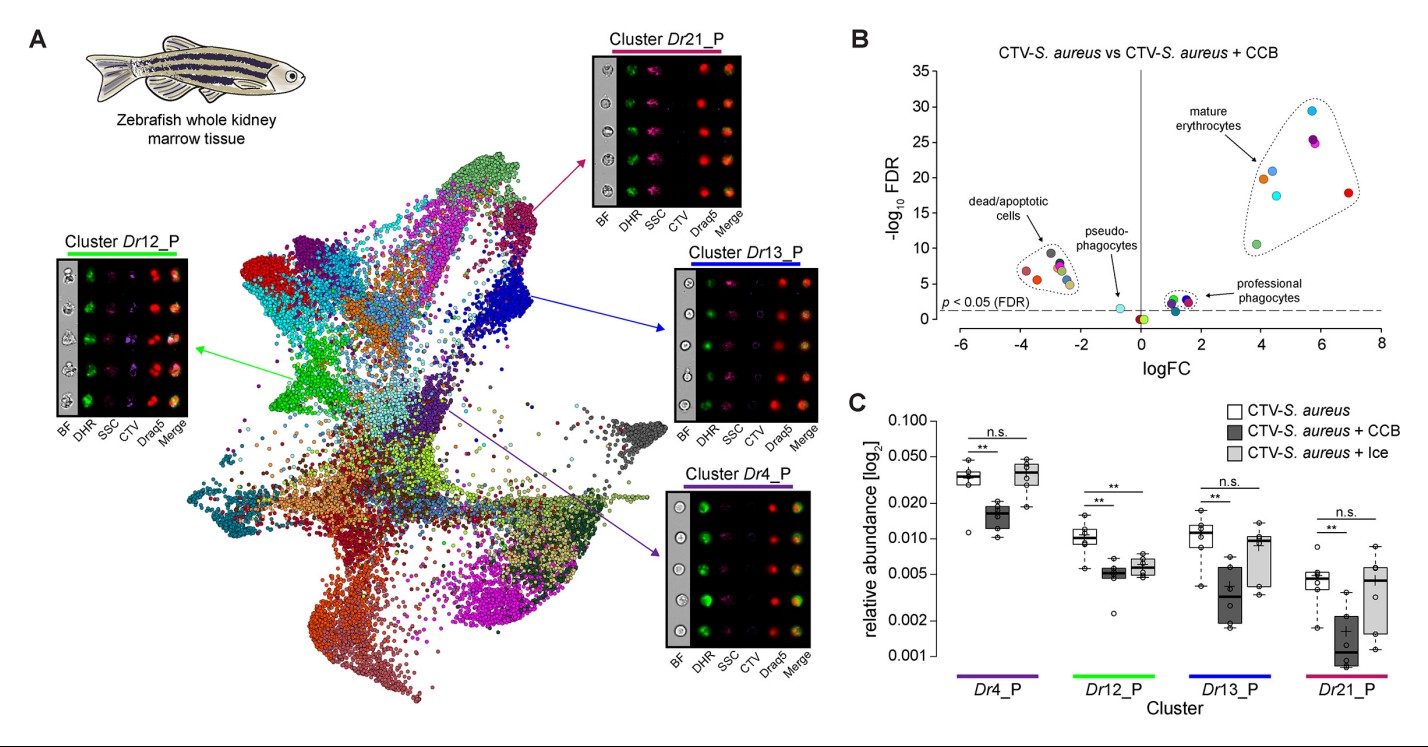

**Figure 3.** Identification of professional phagocytes in zebrafish whole kidney marrow (WKM). (**A**) A phagocytosis assay was performed on a cell suspension obtained from zebrafish WKM tissue and the samples were subsequently run on the ImageStream$^X$ Mark II (n = 6). Force-directed layout (FDL) graph shows 26 clusters, and each color represents a unique cell cluster. Representative cell images belonging to the 4 clusters containing professional phagocytes are shown. Representative cells for all the identified clusters are shown in **Supplementary file 7**. Merge represents the overlay of dihydrorhodamine-123 (DHR) (reactive oxygen species indicator), CellTrace Violet (CTV) (*S. aureus* labeling), and Draq5 (nuclear staining) channels. **Supplementary file 2** reports the features used for this clustering. (**B**) Volcano plot illustrates comparison of cluster relative abundance between phagocytosis samples (CTV-*S. aureus*) and inhibited phagocytosis samples (CTV-*S. aureus* + CCB). The log fold change (logFC) is plotted in relation to the false discovery rate (FDR)-corrected p-value (-log$_{10}$) of each individual cluster calculated with negative binomial regression model (n = 6) (**Supplementary file 5**). Dot color follows the same color code used in (**A**). (**C**) Box plot of relative abundances of events within the 4 clusters containing professional phagocytes in the 3 samples: phagocytosis samples (CTV-*S. aureus*), CCB-inhibited phagocytosis samples (CTV-*S. aureus* + CCB), and ice-inhibited phagocytosis samples (CTV-*S. aureus* + Ice) (**Figure 3—figure supplement 2**). Statistically significant differences are calculated using the negative binomial regression model between the phagocytosis and the inhibited phagocytosis samples (**Supplementary files 5** and **6**). ** indicates p≤0.01 and n.s. indicates not significantly different after FDR (n = 6).

The online version of this article includes the following figure supplement(s) for figure 3:

**Figure supplement 1.** Sample correlation for phagocytosis assay on adult zebrafish whole kidney marrow (WKM).

**Figure supplement 2.** Phagocytosis assay on adult zebrafish whole kidney marrow (WKM) (ice phagocytosis inhibition).

relative abundance of clusters containing dead and apoptotic cells (**Figure 3B**; **Supplementary file 5**). Both outcomes are likely due to reduced cell viability of erythrocytes upon CCB incubation. Moreover, we excluded the possibility of higher cell death in the professional phagocytes upon CCB incubation since pseudo-phagocytes (phagocytes with DHR response but no internalized CTV-*S. aureus*) were significantly more abundant in the CTV-*S. aureus* + CCB sample (**Figure 3B**; **Supplementary file 5**). These results are remarkable since Image3C allowed us to observe a specific effect of CCB on erythrocytes' viability in zebrafish that, as far as we know, was not described before.

Image3C analysis also uncovered another important biological observation. When we inhibited phagocytosis by incubating the single cell suspension on ice (CTV-*S. aureus* + Ice) and compared the specificity of inhibition with the CTV-*S. aureus* + CCB sample (**Figure 3C**; **Supplementary file 6**), we discovered that the inhibition of phagocytosis through low temperature only affects adhesive neutrophils (cluster *Dr*12_P) (**Figure 3C**). This is suspected to occur as ice inhibits adhesion, while CCB effectively blocks phagocytosis in all professional phagocytes in zebrafish WKM tissue by acting on

the cytoskeleton. The use of Image3C allowed us to specifically identify cell types that are sensible to low temperature and those that are not, confirming the existence of different phagocytosis mechanisms and providing additional knowledge about pros and cons of different protocols that can be applied to inhibit phagocytosis based on specific goals and needs.

## Image3C recapitulates cell composition of freshwater snail hemolymph

Since we aimed to provide a tool that is widely applicable, we tested Image3C versatility on the apple snail *Pomacea canaliculata*, an emerging organism for which molecular and cell biological tools have yet to be fully developed. As such, we repeated the same experiments done in zebrafish on the hemolymph of *P. canaliculata.* For morphological examination of the cellular composition of the hemolymph collected from female adults in homeostasis conditions, we stained the single cell suspensions with Draq5 (DNA dye) and ran on the ImageStream$^X$ Mark II. We used Image3C to analyze the images of the events and identified 9 cell clusters (*Figure 4A*, *Figure 4—figure supplement 1*). Two of these clusters comprised cell doublets, debris, and dead cells (clusters *Pc*5 and *Pc*8) and the other clusters, based on inspection of cell images, were grouped into two main categories (*Figure 4A*; *Supplementary file 8*). The first category includes small blast-like cells (cluster *Pc*4) and intermediate cells (clusters *Pc*2 and *Pc*3) with high nuclear-cytoplasmic (N/C) ratio. These cells morphologically resemble the Group I hemocytes previously described using a classical morphological approach (*Accorsi et al., 2013*). The second category comprised larger cells with lower N/C ratio and abundant membrane protrusions (clusters *Pc*1, *Pc*6, *Pc*7, and *Pc*9). Likely, these cells correspond to the previously described Group II hemocytes that include both granular and agranular cells (*Accorsi et al., 2013*).

To identify which of these clusters were enriched for granular cells, we looked at the heatmap with feature values for each individual cluster (*Figure 4B*; see *Supplementary file 1* for feature description). Cluster *Pc*6 had the highest values for the features related to cytoplasm texture and granularity (*i.e.*, area granularity, intensity granularity, and signal granularity) amongst all clusters other than cell doublets (*Figure 4B*; *Supplementary files 3* and *8*). Based on these data, we identified cluster *Pc*6 as the one containing the granular hemocytes. The clusters obtained by Image3C were not only homogeneous and biologically meaningful but were also consistent with published *P. canaliculata* hemocyte classification obtained by classical morphological methods (*Accorsi et al., 2013*). Such remarkable consistency was observed both in terms of identified cell morphologies and their relative abundance in the population of circulating hemocytes (*Figure 4C*; *Supplementary file 8*). For example, the relative abundance of the previously reported small blast-like cells is 14.0%, a value almost identical to the abundance of the corresponding cluster *Pc*4 (13.8%).

Similarly, the category of larger hemocytes or Group II hemocytes represents 80.4% of the circulating cells as measured by traditional morphological methods, while clusters *Pc*1, *Pc*6, *Pc*7, and *Pc*9 combined represent 72.4% of the events analyzed with Image3C (*Figure 4C*; *Supplementary file 3*). A subset of these cells are the granular cells (cluster *Pc*6), which correspond to 7.7% of all hemocytes by classical histological methods and 8.9% by Image3C. The intermediate cells (clusters *Pc*2 and *Pc*3) are the least represented in both approaches, with a relative abundance of 5.6% and 10.6% for the manually and Image3C analyzed events, respectively (*Figure 4C*; *Supplementary file 3*). This difference is likely best explained by the remarkable difference in both the number of cells and the number of features that can be considered for analysis by Image3C. Only a few hundred hemocytes were visually analyzed using traditional histological methods based only on cell diameter and N/C ratio (*Accorsi et al., 2013*). In contrast, the automated pipeline used by Image3C facilitated the analysis of 10,000 nucleated events for each sample and considered 25 morphological features for each cell. The significantly higher number of morphological features simultaneously considered by Image3C also explains the higher number of clusters and improved resolution to distinguish cell types compared to the traditional methods. Hence, Image3C not only can properly analyze cells obtained from an emerging research organism generating biologically meaningful and informative clusters but also represents an unprecedented increase in the accuracy of cell type identification over traditional histological methods, while also allowing high-throughput capability.

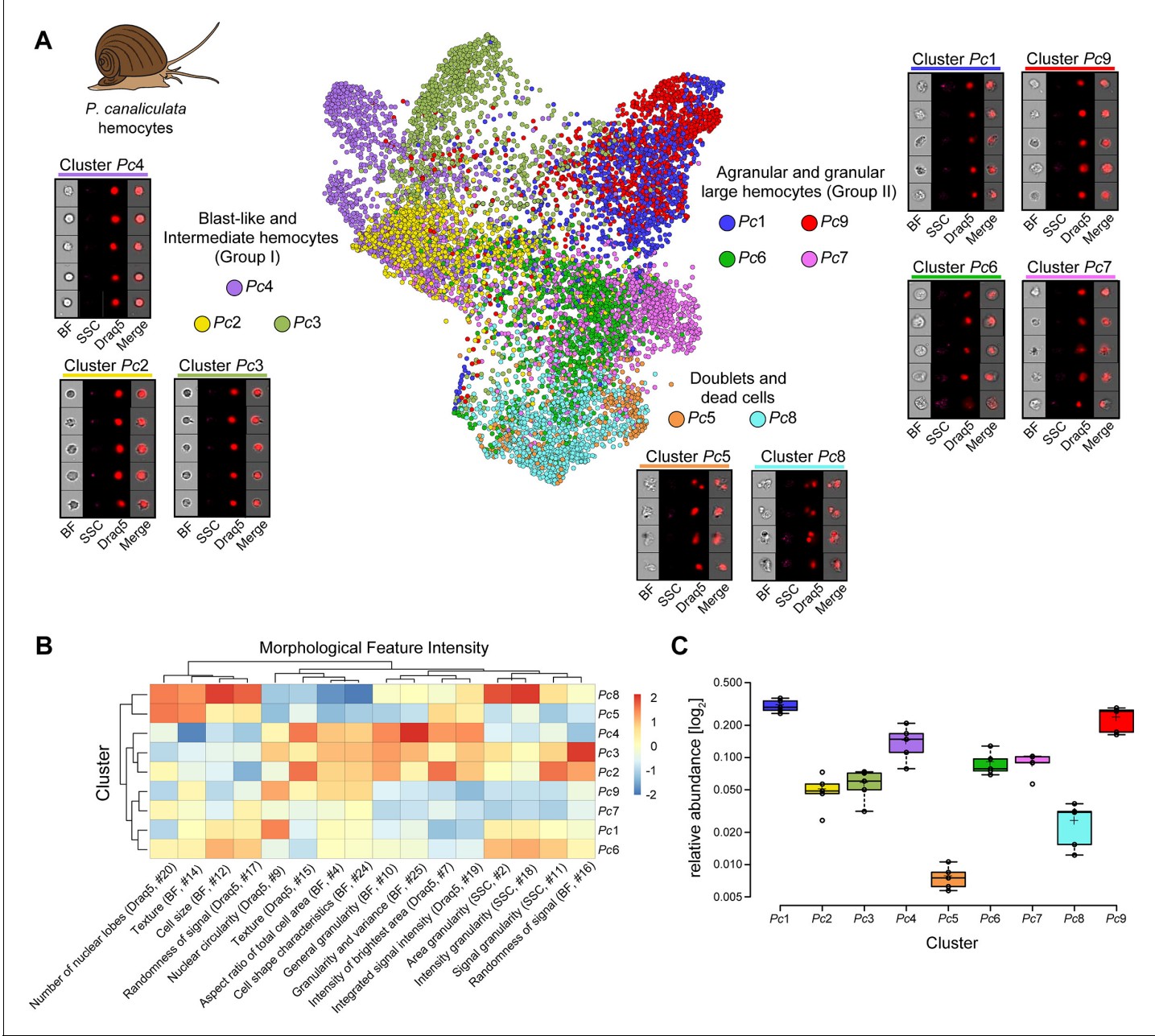

**Figure 4.** Analysis of *P. canaliculata* hemocyte population using only intrinsic morphological features of the cells. (**A**) Hemocytes obtained from the apple snail *P. canaliculata* are prepared for image-based flow cytometric analyses and run on the ImageStream^X Mark II (n = 5). Standard gating of focused and nucleated events was performed using IDEAS software. The selected images were processed through the pipeline described in *Figure 1* and clustered based only on intrinsic morphological and fluorescent feature values. Force-directed layout (FDL) graph is used to visualize the 9 identified clusters, and each color represents a unique cell cluster. Representative cell images belonging to each cluster are shown to evaluate the homogeneity of the cluster and determine morphology of the cells for cluster annotation. Additional representative cells for all the identified clusters are shown in *Supplementary file 8*. Merge represents the overlay of brightfield (BF) and Draq5 (nuclear staining) signal. (**B**) Spearman's correlation plot shows the average feature values of the images in each cluster to highlight morphological similarities and differences between events belonging to different clusters, such as cell size or cytoplasm granularity (*Supplementary file 1*). Cluster *Pc*6 is the one among large hemocytes with higher values in features describing cytoplasm granularity (*i.e.*, area granularity #2, intensity granularity #18, and signal granularity #11). (**C**) Box plot of relative abundance of events within each cluster following the same color code used in (**A**). Clusters *Pc*5 and *Pc*8, constituted by duplets and dead cells, are those with the lowest number of events, validating the protocol used to prepare these samples.

The online version of this article includes the following figure supplement(s) for figure 4:

**Figure supplement 1.** Sample correlation for morphology assay on *P. canaliculata* hemocytes.

# Image3C identifies phagocytosis-competent cells in the hemolymph of a freshwater snail

As with zebrafish, we also performed a phagocytosis experiment on hemocytes from *P. canaliculata*. Our goal was to test if it is possible with an emerging research organism to successfully discover cell phenotypes and functions and obtain information about specific biological processes of interest by using Image3C to compare cell populations among treated and control samples.

Here, we setup the phagocytosis assay incubating the cells with CTV-*S. aureus* and DHR at room temperature. The phagocytosis was inhibited, as control, either adding EDTA (CTV-*S. aureus* + EDTA) or using low temperature by incubating samples on ice (CTV-*S. aureus* + Ice). Events collected on the ImageStream[X] Mark II were analyzed with Image3C and clustered in 20 distinct clusters using quantifications of morphological and fluorescent features (see ***Supplementary file 2*** for feature description), including nuclear staining, phagocytized *S. aureus,* and DHR positivity (***Figure 5A***, ***Figure 5—figure supplement 1***). We compared the phagocytosis-permissive samples (CTV-*S. aureus*) with samples where phagocytosis was inhibited by EDTA incubation or low temperature

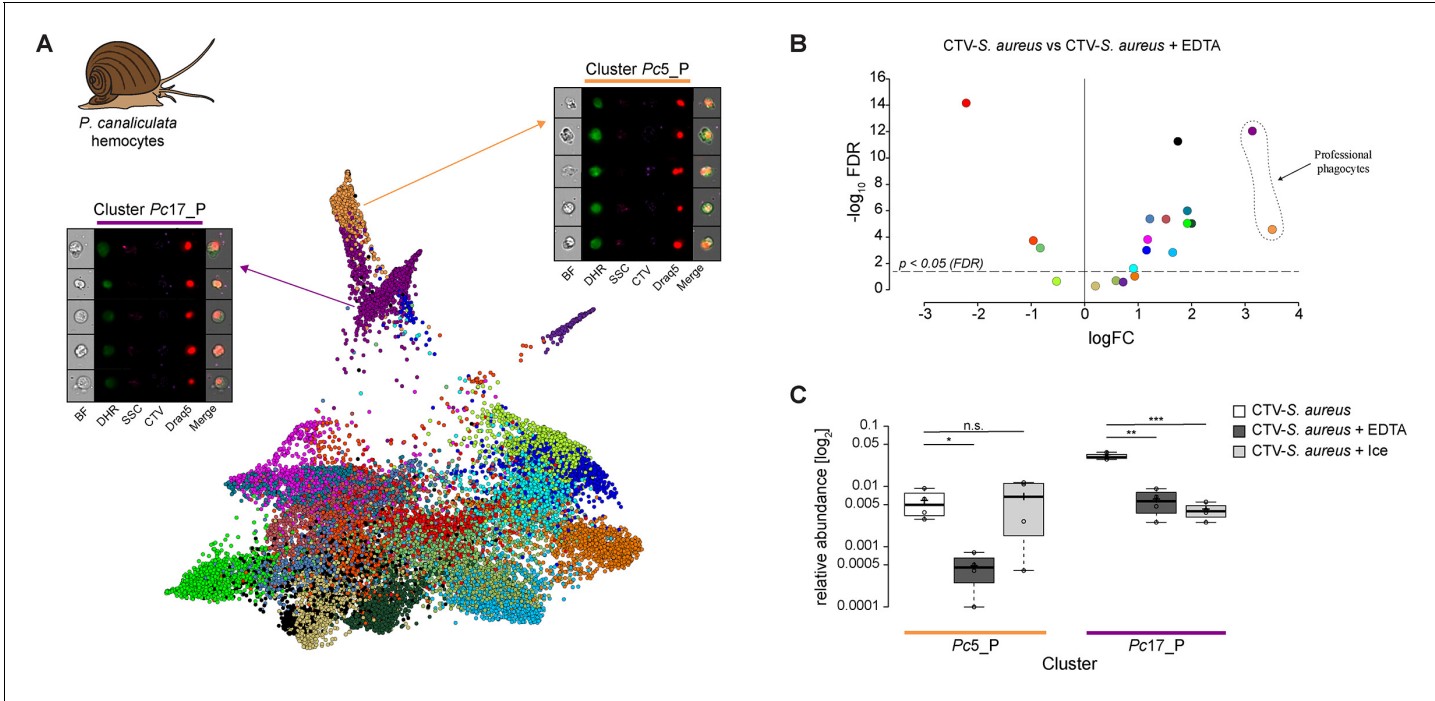

**Figure 5.** Identification of professional phagocytes among *P. canaliculata* hemocytes. (**A**) A phagocytosis assay was performed on a cell suspension obtained from apple snail *P. canaliculata* hemolymph, and the samples were subsequently run on the ImageStream[X] Mark II (n = 5). Force-directed layout (FDL) graph shows 20 clusters, and each color represents a unique cell cluster. Representative cell images belonging to the two clusters containing professional phagocytes are shown. Representative cells for all the identified clusters are shown in ***Supplementary file 11***. Merge represents the overlay of dihydrorhodamine-123 (DHR) (reactive oxygen speciesindicator), CellTrace Violet (CTV) (*S. aureus* labeling), and Draq5 (nuclear staining) channels. ***Supplementary file 2*** reports the features used for this clustering. (**B**) Volcano plot illustrates comparison of cluster relative abundance between phagocytosis samples (CTV-*S. aureus*) and inhibited phagocytosis samples (CTV-*S. aureus* + EDTA). The log fold change (logFC) is plotted in relation to the false discovery rate (FDR)-corrected p-value (-log$_{10}$) of each individual cluster calculated with negative binomial regression model (n = 4) (***Supplementary file 9***). Dot color follows the same color code used in (**A**). (**C**) Box plot of relative abundances of events within the two clusters containing professional phagocytes in the 3 samples: phagocytosis samples (CTV-*S. aureus*), EDTA-inhibited phagocytosis samples (CTV-*S. aureus* + EDTA), and ice-inhibited phagocytosis samples (CTV-*S. aureus* + Ice) (***Figure 5—figure supplement 2***). Statistically significant differences are calculated using the negative binomial regression model between the phagocytosis and the inhibited phagocytosis samples (***Supplementary files 9*** and ***10***). ** indicates p≤0.01 and n.s. indicates not significantly different after FDR (n = 4).

The online version of this article includes the following figure supplement(s) for figure 5:

**Figure supplement 1.** Sample correlation for phagocytosis assay on *P. canaliculata* hemocytes.

**Figure supplement 2.** Phagocytosis assay on *P. canaliculata* hemocytes (ice-inhibited phagocytosis).

**Figure supplement 3.** Feature heatmap after phagocytosis assay on *P. canaliculata* hemocytes.

**Figure supplement 4.** Gallery of *P. canaliculata* hemocytes after phagocytosis assay.

using the statistical analysis included in Image3C based on a negative binomial regression model (*Figure 5B, C*, *Figure 5—figure supplement 2*; *Supplementary files 9* and *10*). The clusters with relative abundance significantly higher in the phagocytosis samples (*Figure 5B*; *Supplementary files 3* and *11*) and with high intensities of both DHR and bacteria signals (*Figure 5—figure supplements 3* and *4*) are the two clusters that we identify as enriched with professional phagocyte (cluster *Pc*5_P and *Pc*17_P) (*Figure 5B*, *Figure 4—figure supplement 1*, *Figure 5—figure supplement 4*; *Supplementary file 11*). The two clusters show a different DHR signal intensity (ROS response) from one another upon bacteria exposure (cluster *Pc*5_P with high DHR signal, cluster *Pc*17_P with low DHR signal) (*Figure 5—figure supplement 3*; *Supplementary files 3* and *11*). Both *Pc*5_P and *Pc*17_P relative abundance is significantly higher in the phagocytosis samples compared to the EDTA-treated sample (*Figure 5C*; *Supplementary file 9*), showing that EDTA effectively inhibits phagocytosis for both types of professional phagocytes. In the sample where the phagocytosis was inhibited by low temperature, however, only cluster *Pc*17_P had a significantly lower relative abundance compared to the phagocytosis sample (*Figure 5C*, *Figure 3—figure supplement 2*; *Supplementary file 10*). We can conclude that similar to CCB inhibition in the zebrafish phagocytosis experiment, EDTA is a more effective and generalized (not cell type-specific) inhibitor of phagocytosis than low temperature. These results show that also in an emerging research organism Image3C allowed discovery of new aspects of this biological process and highlighted differences among professional phagocytes that would have been difficult to detect with other methods.

The data analysis with Image3C clearly highlighted that CCB and EDTA, two classical phagocytic inhibitors commonly used in controls for phagocytosis experiments in vertebrates and invertebrates, respectively, result in a drastic change of cell morphology and cell viability. This consequence is not easily detectable by other methods and is therefore often overlooked. In the present work, these changes significantly modified the overall cell cluster number and distribution and indicate that the effects of CCB and EDTA on cell morphology should be taken into consideration in any study of morphological features of cells with phagocytosis properties because artifacts might be significant.

## CNN allows unbiased comparison between experiments

When determining differences between control and experimental treatments, Image3C necessarily combines images and data from all samples and then clusters the cells. This must be taken into consideration for experimental planning. Experiments meant to analyze cell composition and morphological diversity in one biological domain (*e.g.*, homeostasis condition) (*Figure 2*, *Figure 4*) should be carried out separately from those in other domains that are likely to introduce changes in the cell population composition or cell morphologies representing a confounding factor for the *de novo* clustering in homeostasis condition. Image3C clustering works best when used, at the same time, only on samples belonging to a single experimental domain, such as homeostasis or the phagocytosis assay. An issue that emerges when analyzing different experimental sets independently is the increase of time requirement for analytical steps, the likelihood of introducing errors, and the need to repeatedly annotate the clusters in the FDL graph obtained from each experimental set. This last element is required for comparing cell type behaviors among multiple experiments and provide a global understanding of their functions and response to treatments (*i.e.*, cluster #1 from one analytical run cannot be expected to match cell morphologies with cluster #1 from another run since there is a stochastic element to the process).

This last point is probably the most challenging since mistakes can easily be introduced based on user biases or lack of sufficient pre-existing knowledge about cell morphologies or of cell biomarkers that would allow a confident cross-annotation between multiple FDL graphs. In addition, we observed that the number of clusters drastically increases when including treatments that influence cell morphological properties of the cell. As an example, while we detected 9 unique clusters in naïve hemolymph samples, we detected 20 clusters in the phagocytosis experiment (*Figure 3A*, *Figure 4A*). This is in part due to the fact that professional phagocytes change their morphology upon detection of pathogens (*Palić et al., 2007*), thus creating new clusters. Similarly, the complexity of the clustering is also increased by treatments, such as CCB and EDTA incubations, that are necessary to ensure identification of professional phagocytes, but have a strong impact on the morphology of the cells making the clustering and annotation steps more challenging and prone to mistakes since treated samples contain aberrant populations not found at homeostasis (*Figure 5A*; *Supplementary file 11*).

To provide an alternative for streamlining the analysis of multiple experimental sets upon initial *de novo* clustering and cell type identification in homeostasis samples, we included in Image3C the possibility to use these initial images and their cluster IDs to train a CNN without manually classifying the images (*Figure 1*). This trained classifier can then be used to assign the cell images subsequently collected from additional experimental sets to one of the clusters defined in the homeostasis condition in a high-throughput way. In this way, it will be possible to determine the behavior of a specific cell type through multiple experimental sets without re-clustering whenever new data is acquired. A crucial element that allows this approach is also represented by the ImageStream$^X$ Mark II system that provides highly reproducible and comparable images of cells coming from different experiments and acquired at different days, introducing much less variability than standard light or electron microscopy.

For our pipeline, then, a CNN (*LeCun et al., 1989*) based on the architecture of DenseNet (*Huang et al., 2017*) was deployed to (1) use, as training set, images and clusters obtained from a first group of samples (*e.g.*, homeostasis conditions, naïve cells, or WT samples) analyzed in an unbiased way by *de novo* clustering and (2) assign new cell images acquired through ImageStream$^X$ Mark II system to their respective classes. As proof of concept, we used the clusters identified for *P. canaliculata* hemocytes in homeostasis condition with the first part of the pipeline (*Figure 4A*) for training and setting up the CNN classifier. This approach would define the classes based on the unbiased *de novo* clustering of thousands of cells with no need for formal annotation or previous knowledge about cell types and tissue composition. To prepare a dataset for training the classifier, we first combined clusters that strongly overlapped with one another in terms of morphological characteristics (*e.g.*, doublets and dead cells) to increase accuracy of the classifier (*Figure 6A*). We used 80% of the cells obtained in the original *P. canaliculata* dataset together with their cluster IDs to train the classifier through over 25,000 iterations. After each iteration, we tested the training with 10% of the original dataset and determined the relative accuracy by scoring numbers of cells whose cluster ID assigned by the classifier matched the original cluster ID (*Figure 6B, C*). The remaining 10% of the original dataset was used to calculate the precision of the trained classifier. Clusters with higher support numbers obtained higher precision scores. The weighted average precision score (f1-score, precision average score across clusters controlling for support numbers) of 0.75 is relatively high considering the complexity of the phenotype (brightfield [BF], darkfield, and Draq5 images) (*Figure 6D*) and comparable to other studies using ML for cell classification (*Blasi et al., 2016*). The true probability match for each individual cell (probability for each cell that the class assigned by the classifier would match the original cluster ID) demonstrated that lower true probability matches occurred where clusters strongly overlapped or where cell phenotypes are intermediate between clusters, providing an additional layer of information about our dataset (*Figure 6D*).

To test the efficiency of this pipeline, we extracted all the images belonging to the two clusters identified in the phagocytosis assay as cluster-containing phagocytes and determined to which naïve cell type they correspond using the CNN classifier and only the BF, SSC, and Draq5 channels (*i.e.*, DHR and labeled bacteria signals were not used). We found that 59.4, 6.2, and 9.2% of the phagocytes belonged to cluster *Pc*1_CT, *Pc*6_CT, and *Pc*7_CT, respectively (*Figure 6E*), where CT stands for classifier training. These results confirmed a previously published result that used classical morphological staining and manual annotation to conclude that the hemocytes able to phagocytize were primarily Group II hemocytes (*Accorsi et al., 2013*). Only 8% of the phagocytes were clustered in the Group I hemocytes, here represented by clusters *Pc*2_CT, *Pc*3_CT, and *Pc*4_CT, while the remaining 17.2% were assigned by the CNN to the cluster *Pc*5_CT, constituted by doublets and dead cells (*Figure 6E*). This result can be explained by the fact that *in vitro* phagocytosis triggers microaggregate formation (hemocyte-hemocyte adhesion) in invertebrate hemocytes that resembles the nodule formation observed *in vivo* (*Walters, 1970*). It is important to observe how this analysis allowed us to assign phagocytes to cell types using the annotation already performed in *Figure 4A* (*de novo* clustering of hemocytes in homeostasis condition) without the need to reannotate the FDL obtained during the phagocytosis assay (*Figure 5A*).

To test the adaptability of the trained CNN to new datasets, we collected hemocytes from male apple snail specimens, stained the cells with Draq5, and recorded BF, SSC, and nuclei images from 10,000 cells on the ImageStream$^X$ Mark II as previously described. We extracted the images of the cells and used the trained CNN classifier to determine the relative abundance of hemocytes collected from male snails in the seven classes of the classifier (*Figure 6F*). First, we visually compared

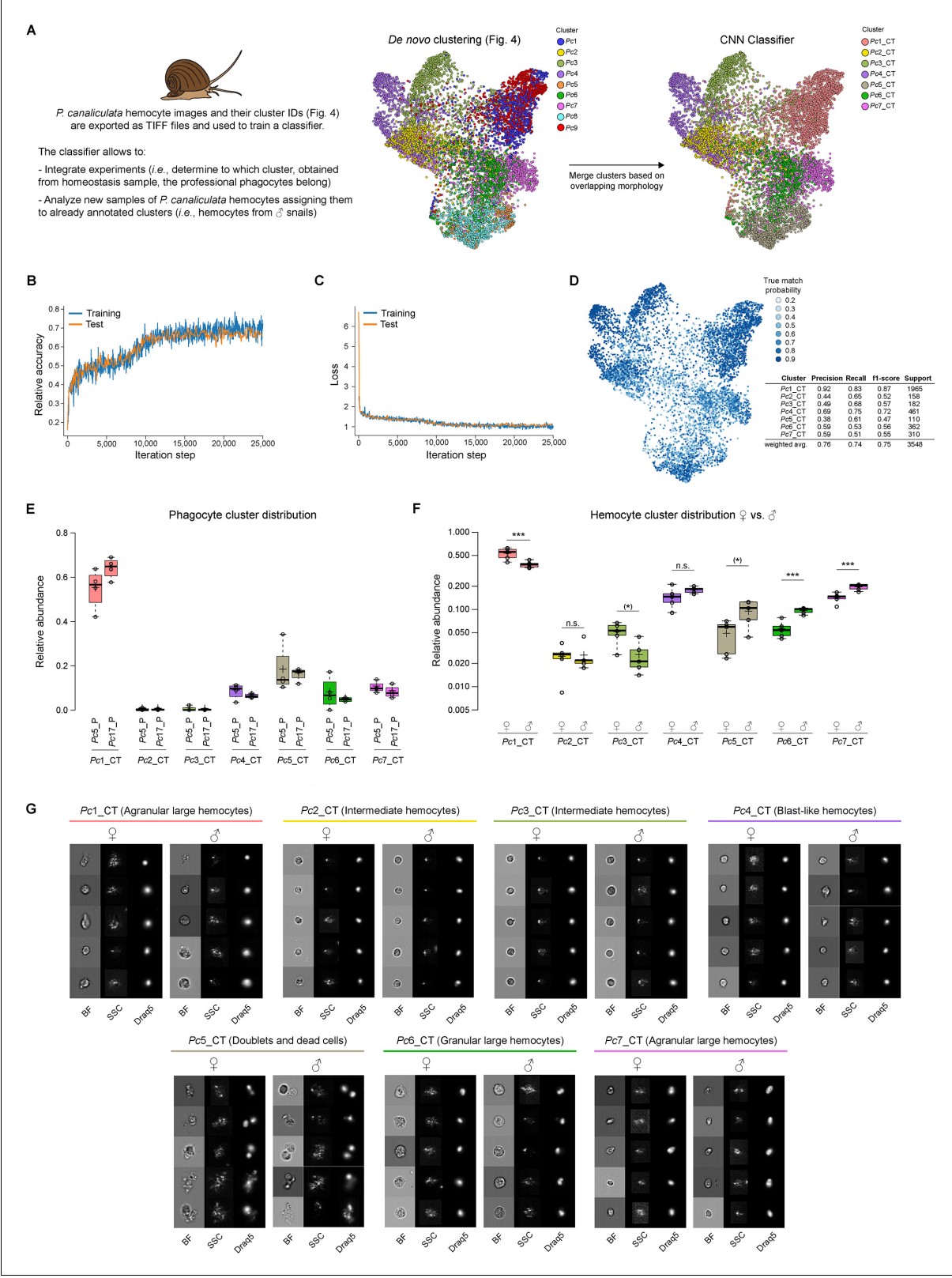

**Figure 6.** Classifier training and use of convolutional neural network (CNN) for integrating multiple experiments. (**A**) The CNN portion of Image3C allows the integration of multiple experiments and the analysis of additional samples assigning new events to already defined and annotated clusters without manual and potentially biased matching of clusters by the user. Images of cells obtained from homeostasis, naïve, or WT samples and already *de novo* clustered in an unbiased way by Image3C are exported as TIFF files together with their cluster IDs. *P. canaliculata* hemocytes in homeostasis

*Figure 6 continued on next page*

*Figure 6 continued*

condition obtained from female apple snails (35,000 images) is the dataset used for training the CNN classifier. Before the training, clusters with cells with strongly overlapping morphology were merged (*Pc*1 was merged with *Pc*9 and named *Pc*1_CT; *Pc*5 was merged with *Pc*8 and named *Pc*5_CT). CT: classifier training. (B) The training was performed on 80% of the exported images, and the testing during the training was performed on 10% of the exported images. Relative accuracy was recorded every 100 iterations for 25,000 iterations total. (C) Loss calculation was recorded every 100 iterations for 25,000 iterations total and indicates that the training set is not memorized. (D) The True match probability (probability that trained classifier-assigned cluster matches original cluster ID) is given for each cell of the remaining 10% of the original exported images. The detailed precision score for each cluster together with the weighted average (correcting for support) is reported in the table. (E) The CNN classifier allows for integrating experiments, such as phagocytosis assay and morphological assay, with minimal probability of introducing errors because of lack of biomarkers. The plot shows the distribution of the cells belonging to the snail professional phagocytes clusters (*Pc*5_P and *Pc*17_P from **Figure 5**, phagocytosis assay) among the clusters defined by morphological features using homeostasis conditions. (F) The CNN classifier allows also for the analysis of new samples obtained from the same species and the same tissue used for the training. The new events obtained running male *P. canaliculata* hemocytes are assigned to clusters defined by the *de novo* clustering step, allowing for comparison between samples (females vs. males) and statistical analysis for cluster relative abundance differences. The box plot shows the comparison of the hemocyte population composition between females and males (n = 5). *** are clusters with abundances robustly significantly different between female and male animals. * are clusters with p-value lower than 0.05, but whose abundances are no more significantly different after the correction for multiple testing (false discovery rate [FDR]). (G) Representative cell images for each cluster belonging to female and male hemocytes dataset allow to visually compare the cells used to train the classifier (female) with cells assigned by the classifier and coming from a new set of samples (males). Brightfield (BF), side scatter signal (SSC), and Draq5 (nuclear staining) signal are shown in individual channels.

the female hemocytes clustered by the *de novo* clustering with the male hemocytes that were run on the ImageStream^X Mark II and were assigned to a class by the classifier (**Figure 6F**). This comparison shows that the female and male hemocytes belonging to the same cluster are morphologically extremely similar and different from the hemocytes assigned to other clusters (**Figure 6F**). This demonstrates that the CNN classifier can be trained with a first group of samples and then it can successfully analyze new datasets acquired later on. The comparison between female and male hemocyte compositions revealed that the clusters significantly different in terms of relative abundance are *Pc*1_CT and *Pc*7_CT (Group II agranular large hemocytes) and *Pc*6_CT (Group II granular large hemocytes) (**Figure 6F**). Significantly, prior studies detected no differences between females and males hemocytes composition through manual classification and counting using a classical morphological approach (*Accorsi et al., 2013*). The reduced user bias and high-throughput analysis presented here, in contrast, allowed us to determine that one of the two subpopulations of agranular large hemocytes was significantly more abundant in the female animals (*Pc*1_CT: 53 and 38% in females and males, respectively) while the other agranular (*Pc*7_CT) as well as the granular large hemocytes (*Pc*6_CT) was significantly more abundant in the male animals (*Pc*7_CT: 14 and 20% in females and males, respectively; *Pc*6_CT: 6 and 10% in females and males, respectively) (**Figure 6F**).

While the biological significance of this observation is not going to be further investigated in this paper, the discovery highlights the power of Image3C analysis compared to traditional methods for determining and quantifying the composition of cell populations. These experiments demonstrate that Image3C, in combination with the presented convolutional classifier, can analyze large experimental datasets and identify significances with small effect sizes. Importantly, Image3C analysis is independent of observer biases and does not require prior knowledge about expected tissue composition or the expected effect of treatment on cell morphology.

## Conclusion

We have developed a powerful new method to analyze at single-cell resolution the composition of any cell population obtained from research organisms for which species-specific reagents (such as fluorescently tagged antibodies), biomarkers, single-cell atlases, or a high-quality genome for a scRNA-seq approach are not available. We demonstrated that Image3C can identify different cell populations based on morphology and/or function through *de novo* clustering and highlight important changes in cell type abundance and cell population composition caused by experimental or natural perturbation (sex, treatment, experimental protocol). Image3C does not require, at any point, prior knowledge about the tissue composition or cell type-specific markers, although, if available, they can be included and used. Furthermore, in combination with the CNN classifier trained on these clusters, we demonstrate that Image3C is capable of bias-free and high-throughput analysis of large experimental datasets making it possible to compare a specific cell type behavior or population

composition across multiple experiments. Image3C is extremely versatile and can be applied to any tissue or cell population of interest and is adaptable to a variety of experimental designs. Although Image3C was developed in response to the need of analyzing cell composition of tissues in emerging research organisms, the Image3C tool could be potentially used also to add to transcriptomic dataset an additional and complementary layer of information based on cell morphology. Given the recent advancement in image-based flow cytometry that enables image capturing together with cell sorting (*Nitta et al., 2018*), a scRNA-seq approach in combination with the Image3C pipeline would enable simultaneous analysis of both the morphological/phenotypic and genetic properties of a cell population at single-cell resolution.

# Materials and methods

## Key resources table

| Reagent type (species) or resource | Designation | Source or reference | Identifiers | Additional information |
|---|---|---|---|---|
| Strain, strain background (*Staphylococcus aureus*) | Wood strain without protein A | Thermo Fisher Scientific | S2859 | |
| Biological sample (*Danio rerio*) | Whole kidney marrow | Stowers Institute for Medical Research | Wild type, adult females | Freshly isolated from *Danio rerio* |
| Biological sample (*Pomacea canaliculata*) | Hemolymph | Stowers Institute for Medical Research | Wild type, adult females and males | Freshly isolated from *Pomacea canaliculata* |
| Chemical compound, drug | Dihydrorhodamine-123 (DHR) | Thermo Fisher Scientific | D23806 | 5 µM |
| Software, algorithm | IDEAS | Amnis Millipore Sigma | Version 6.2 | |
| Software, algorithm | R code | This paper | | https://github.com/ stowersinstitute/ LIBPB-1390-Image3C (*Box, 2021*) |
| Software, algorithm | VorteX clustering environment | https://github.com/ nolanlab/vortex/ releases | | |
| Software, algorithm | FSC Express | De Novo Software | Image or Plus configurations - Version 7 | |
| Software, algorithm | Python script | This paper | | https://github. com/stowersinstitute/ LIBPB-1390-Image3C |
| Other | Draq5 | Thermo Fisher Scientific | 62251 | 5 µM |
| Other | CellTrace Violet (CTV) | Thermo Fisher Scientific | C34571 | 5 µM |

## Collection of zebrafish WKM

Twelve-month-old, wild type, female, adult zebrafish were euthanized with cold 500 mg/L MS-222 solution for 5 min. WKM was dissected as previously described (*Traver et al., 2003*) and transferred to 40 µm cell strainer with 1 mL of L-15 media containing 10% water, 10 mM HEPES, and 20 U/mL Heparin (L-90). Cells were gently forced through the cell strainer with the plunger of a 3 mL disposable syringe. The strainer was washed once with 1 mL of L-90 and the resulting single cell suspension was centrifuged at 500 rcf at 4 °C for 5 min. The supernatant was discarded, and the cells were resuspended in 1 mL of L-90 containing 5% fetal calf serum (FCS), 4 mM L-glutamine, and 10,000 U of both penicillin and streptomycin (L-90 media). The cells were counted in a 1:20 dilution on the EC-800 flow cytometer (Sony) using scatter properties.

## Collection of apple snail hemocytes

Specimens of the apple snail *P. canaliculata* (Mollusca, Gastropoda, Ampullariidae) were maintained and bred in captivity, in a water recirculation system filled with artificial freshwater (2.7 mM $CaCl_2$, 0.8 mM $MgSO_4$, 1.8 mM $NaHCO_3$, 1:5000 Remineralize Balanced Minerals in Liquid Form [Brightwell Aquatics]). The snails were fed twice a week and kept in a 10:14 light:dark cycle. Wild-type adult snails, 7–9 months old and with a shell size of 45–60 mm, were starved for 5 days before the hemolymph collection (*Accorsi et al., 2013*). If not differently specified, female snails were used for the experiments. The withdrawal was performed by applying a pressure on the operculum and dropping the hemolymph directly into an ice-cold tube. The hemolymph collected from different animals was not pooled together. The hemolymph was immediately diluted 1:4 in Bge medium + 10% fetal bovine serum (FBS) and then centrifuged at 500 rcf for 5 min. The pellet of cells was resuspended in 100 µL of Bge medium + 10% FBS. The Bge medium (also known as *Biomphalaria glabrata* embryonic cell line medium) is constituted by 22% (v/v) Schneider's *Drosophila* Medium, 4.5 g/L lactalbumin hydrolysate, 1.3 g/L galactose, 0.02 g/L gentamicin in MilliQ water, pH 7.0.

## Experiment 1: morphology assay in homeostasis conditions

WKM cells from zebrafish were isolated as described before and plated at $4 \times 10^5$ cells/well in a 96-well plate in 200 µL of L-90 media and incubated for 3 h at room temperature. Cells were stained with 5 µM Draq5 (Thermo Fisher Scientific) for 10 min and subsequently run on the ImageStream$^X$ Mark II (Amnis Millipore Sigma), where 10,000 nucleated and focused events were recorded for each sample (n = 8). Erythrocytes were outgated to enrich for immune-relevant cells and prevent overclustering in the subsequent analysis. The latter is due to the fact that fish erythrocytes are nucleated and their biconcave shape results in different morphological feature values only depending on their orientation during image acquisition.

The *P. canaliculata* hemocytes were stained with 5 µM Draq5 for 10 min, moved to ice, and subsequently run on the ImageStream$^X$ Mark II, where 10,000 nucleated and focused events were imaged for each sample (n = 5).

## Experiment 2: phagocytosis assay

*Staphylococcus aureus* (Thermo Fisher Scientific) were resuspended in PBS at the final concentration of 100 mg/mL and incubated with 5 µM CTV (Thermo Fisher Scientific) for 20 min. Labeled bacteria were centrifuged and resuspended in PBS for three times to remove unbound dye and then stored at −20 ℃ as single-use aliquots. Cells, obtained from fish WKM or snail hemolymph and in a single cell suspension, were plated in a 96-well plate at a concentration of $4 \times 10^5$ cells/well in 200 µL of medium and incubated with $2 \times 10^7$ CTV-coupled *S. aureus*/well for 3 h at room temperature. As control, the phagocytosis was inhibited incubating the cells + CTV-*S. aureus* mix either on ice (for both species) or with 0.08 mg/mL CCB for zebrafish cells or with 30 mM EDTA and 10 mM HEPES for apple snail cells (*Cueto et al., 2015*; *Li et al., 2006*). After 2 h and 30 min, we added 5 µM DHR (Thermo Fisher Scientific) to the cell suspension to stain cells positive for ROS production. To control for this treatment with DHR, we incubated one aliquot of cells with 10 ng/mL phorbol 12-myristate 13-acetate (PMA) to induce ROS production. At 2 h and 50 min since the beginning of incubation with CTV-*S. aureus*, all the samples were stained with 5 µM Draq5 for 10 min. After 3 h incubation with bacteria, cells were moved and stored on ice and subsequently run on the ImageStream$^X$ Mark II, where 10,000 nucleated and focused events were imaged for each sample (at least n = 4 snail and n = 6 fish) at a speed of 1,000 images/s.

## Data collection on ImageStream$^X$ Mark II

Following cell preparation, data were acquired from each sample on the ImageStream$^X$ Mark II at $60\times$ magnification, slow/sensitive flow speed (1,000 images/s), using 633, 488, and 405 nm laser excitation. BF was acquired on channels 1 and 9, DHR (488 nm excitation) on channel 2, CTV-*S. aureus* (405 nm excitation) on channel 7, Draq5 (633 nm excitation) on channel 11, and SSC was acquired on channel 6. Single-color controls were also acquired for each fluorescent channel to allow for fluorescence spillover correction.

## Data analysis and *de novo* cluster identification

An interactive map representing the pipeline, the software used, the format of the exported files, and an approximation of time required for running the individual steps is provided in *Figure 1—figure supplement 1*. Raw images from the ImageStream[X] Mark II system (RIF files, a type of modified 16-bit TIFF file) were compensated (spillover and other corrections applied), background was subtracted, and features were calculated using IDEAS 6.2 software (Amnis Millipore, free for download once an Amnis user account is created). The resulting compensated image files (CIF files) were used to quantify features for all cells and samples. *Supplementary files 1* and *2* report the list of features used for each organism and for each experiment and their description. These per-object feature matrices (DAF files) were then exported from IDEAS into FCS files. Exported FCS files were processed in R (*R Development Core Team, 2014*). In order to trim redundant features that contribute noise but little new information, Spearman's correlation values for each pair of features were calculated using all events of a representative sample and one of the features of the pair was trimmed when correlation between the two was $\geq$0.85 (*Figure 2—figure supplement 1*; *Caicedo et al., 2017*). The Spearman's correlation of the mean values of remaining features per each sample was then used to identify outliers among sample replicates. Samples with correlation of mean feature values below 0.85 with the set were discarded (*Figure 2—figure supplement 2*, *Figure 3—figure supplement 1*, *Figure 4—figure supplement 1*, *Figure 5—figure supplement 1*), although in general the replicates were consistent. Also, while morphological features did not require any transformation or normalization, fluorescence intensity features were transformed using the estimateLogicle() and transform() functions from the R flowCore package (*Ellis et al., 2018*; *Hahne et al., 2009*) to improve homoscedasticity (homogeneity of variance) of distributions. DNA intensity features were also normalized to align all 2N and 4N peak positions and remove intensity drift between samples (*Figure 2—figure supplement 3*) using the gaussNorm() function from flowStats package (*Hahne et al., 2018*). The processed data was exported from R (*R Development Core Team, 2014*) using writeflowSet() function in flowCore package (*Ellis et al., 2018*; *Hahne et al., 2009*) as CSV or FCS files, depending on downstream needs for the file output.

These processed data files were then imported into the VorteX clustering environment for X-Shift k-nearest-neighbor clustering (free to install) (*Samusik et al., 2016*). X-Shift was selected as a clustering method for Image3C based on a previously published analysis and comparison of clustering methods (*Weber and Robinson, 2016*). From that work, we determined that X-shift represents an optimal trade-off: identifying low-frequency populations, accurately identifying 'true' clusters (*i.e.*, F1 scores), not requiring for *a priori* knowledge of the number of clusters (populations) and having reasonable runtimes (due to hardware CPU requirements). We also did an early comparison of X-shift with K-means (data not shown) and determined that K-means was insufficient for our purposes as known cell populations were not well represented by clusters, and we did not want to specify the number of expected clusters into the method's input parameters since this would not be known in experimental use. During the import into VorteX, all features were scaled to 1 SD to equalize the contribution of features towards clustering. Clustering was performed in VorteX testing a range of k values (typically from 5 to 150), choosing a final k value using the 'find elbow point for cluster number' function in VorteX and confirming visually that over- or underclustering did not occur. FDL graphs of a subset of cells obtained from each set of samples were also generated in VorteX, and cell coordinates in the resultant 2D space were exported along with graphML representation of the FDL graph. Finally, tabular data (CSV files) was exported from VorteX including a master table of every cell event with its cluster assignment and original sample ID, as well as a table of the average feature values for each cluster and counts of cells per cluster and per sample.

Clustering results were further analyzed and plotted in R (*R Development Core Team, 2014*) by merging all cell events and feature values with cluster assignments and X/Y coordinates for FDL graph. Using this merged data and the graphML file exported from VorteX, new FDL graphs were created for each treatment condition using the igraph package (*Csardi and Nepusz, 2006*) in R (*R Development Core Team, 2014*). Statistical analysis of differences in cell counts per cluster by condition was performed using negative binomial regression of cell counts per cluster, plots of statistic results and other results were generated, and CSV files containing cell ID, sample ID, feature values, and X/Y coordinates in FDL graph were exported for each sample. The subsequent use of FCS Express Plus version 6 (DeNovo software, free alternative are mentioned later in the text)

allowed visualization of cell images using DAF/CIF files by cluster and customized subsets of the FDL graphs.

DAF files were opened in FCS Express Plus software, and the 'R add parameters' transformation feature with a custom script was used to merge the clustering data saved in the CSV files generated above with both DAF and CIF files (feature values and image sets, respectively). FCS Express Plus was utilized at this stage of work because it is the only platform currently available that works with Amnis DAF and CIF files while also running transformation processes driven by R scripting. ImageJ Bio-Formats allows reading images from DAF and CIF files, but we got pixels with a value much higher than expected, probably due to a bug that has not been fixed yet. This allowed to visualize image galleries of cells within each cluster and gate by features of interest on 2D plots (more traditional flow cytometry analysis) for exploring the clustering results and identifying clusters and populations of interest. FCS Express Plus is a proprietary software, but similar results can be obtained with IDEAS software where a text file with cluster IDs for each image event can be imported and the cluster information can be matched to the event images.

The full complement of R packages used includes flowCore (*Ellis et al., 2018*; *Hahne et al., 2009*), flowStats (*Hahne et al., 2018*), igraph (*Csardi and Nepusz, 2006*), ggcyto (*Jiang, 2015*), ggridges (*Wilke, 2018*), ggplot2 (*Wickham, 2016*), stringr (*Wickham, 2010*), hmisc (*Harrell and Dupont, 2019*), and caret (*Kuhn, 2008*). *Figure 1—figure supplement 1* can be used as an interactive map of Image3C pipeline, where, upon clicking on the different portions of the pipeline, the users will be automatically directed to the corresponding sections of our GitHub repository. The GitHub repository at https://github.com/stowersinstitute/LIBPB-1390-Image3C reports a detailed description of all these processing steps and includes sample scripts, workflow files, and example datasets and tutorials.

## Setup and use of a CNN classifier

Once the clusters were defined with the previously described *de novo* clustering analysis, we used a CNN (*LeCun et al., 1989*) based on the architecture of DenseNet (*Huang et al., 2017*) for image classification. Out of all the events captured with the ImageStream$^X$ Mark II system, we selected only single nucleated objects applying gates on area vs. aspect ratio plot and Draq5 intensity plot to achieve this selection, respectively. For these objects, we exported 16-bit TIFF images (one channel per fluorescence/BF image 'color') using IDEAS 6.2 software.

Because images from the ImageStream$^X$ Mark II system have nonuniform sizes, each image was cropped or padded to $32 \times 32$ pixels using NumPy indexing (*van der Walt et al., 2011*) in a Python script. The CNN consists of three dense blocks that transition from three-channel image input of $32 \times 32 \times 3$ to a final size of $4 \times 4 \times 87$ with 87 feature maps. A dense block includes three convolution layers, each followed by leaky ReLU activation. The last step of the block is a strided convolution used to down-sample the width and height of the feature maps by a factor of 2. The final layer of the CNN flattens the $4 \times 4 \times 87$ array into a 1D vector of length 1392 and is fully connected to the output layer, that is, a 1D vector with a length of the number of classes for prediction. The CNN used softmax cross-entropy for the loss function with L2 regularization, and the Adam optimizer (*Kingma and Ba, 2014*) was used to minimize the loss function. The CNN was implemented in Python using the TensorFlow platform (*Abadi et al., 2015*) and the SciPy ecosystem (*Oliphant, 2006*; *Oliphant, 2007*; *Pedregosa et al., 2011*).

The CNN was used to train a classifier using over 35,000 images of *P. canaliculata* hemolymph cell types in homeostasis condition acquired with the ImageStream$^X$ Mark II system. The event images were randomly split in 80% for training, 10% for testing during training, and 10% for final validation. The first 80% of the images were used together with their cluster IDs obtained by the *de novo* clustering to train the classifier. The learning rate for the Adam optimizer was set to 0.0006 with a decaying learning rate starting at 0.001 and decreasing by 1% each step. The training proceeded for 25,000 iterations with a size of 256 randomly selected images for each iteration. After each iteration, 10% of the cells of the original *P. canaliculata* dataset was used to test the classifier. The loss and accuracy of the CNN were recorded after every 100 iterations to monitor the performance. The CNN loss was defined by the softmax of the cross-entropy (*Dahal, 2017*) between the final output and the one-hot-encoded image labels. To avoid the CNN memorizing the training set, L2 regularization was applied to the weights. The training and test sets follow the same accuracy

and loss trends over all iterations, indicating the training set is not memorized and can generalize to predict the test set.

The finally trained classifier was tested on the remaining 10% of images that were completely new for the CNN. The trained model was saved for future use, so new images can be inferred by the network to predict cell types. The inference is very fast because only one forward pass is made through the network and no backpropagation occurs. The result of the inference is a vector with length equal to the number of cell type classes. Each element of the vector will be the probability of the cell belonging to the corresponding class; the sum of the vector must be 1. Inferring a complete experiment will provide a probability vector for each image; the list of probability vectors can be saved as CSV file. For new images, the inference results will need to be examined to ensure the predictions are reliable. A large majority of the probability vectors should have a maximum greater than 0.5, and a subset of the images should be visually inspected to verify proper class assignment.

The interactive map of Image3C pipeline (*Figure 1—figure supplement 1*) includes also the training and the use of the CNN. The GitHub repository at https://github.com/stowersinstitute/LIBPB-1390-Image3C reports a detailed description of all these processing steps, includes sample scripts, workflow files, and example datasets and tutorials and can be easily accessed by clicking on the right side of the Image3C interactive map (*Figure 1—figure supplement 1*).

## Statistical analysis

Negative binomial regression was performed on tables of cell counts per cluster and per sample and plots were generated using R (*R Development Core Team, 2014*) with the edgeR package (*Robinson et al., 2010*), which was developed for RNA-seq analysis, but includes generally applicable and user-friendly wrappers for regression, modeling analysis, and plotting of results. For the comparison of cellular hemolymph composition between females and males of *P. canaliculata,* a one-way ANOVA with subsequent FDR (Benjamini–Hochberg, correction for multiple testing) was used.

## Acknowledgements

We acknowledge Hua Li for her assistance on the statistical analysis and also thank the Laboratory Animal Services and the Aquatics Facility at the Stowers Institute for Medical Research for animal husbandry. We would like to thank Blair Benham-Pyle, Carolyn Brewster, Viraj Doddihal, Julia Peloggia de Castro, and Barbara Milutinović for their critical comments on an earlier version of this manuscript. This work was supported by institutional funding to ACB, CW, ASA, and NR. ASA is a Howard Hughes Medical Institute Investigator. NR is further supported by the Edward Mallinckrodt Foundation, NIH Grant R01 GM127872, DP2AG071466, and NSF EDGE award 1923372. AA was supported by the Emerging Models grant from the Society for Developmental Biology (SDB) and the postdoctoral fellowship from the American Association of Anatomists (AAA). RP was supported by a grant from the Deutsche Forschungsgemeinschaft (PE 2807/1-1).

## Additional information

### Funding

| Funder | Grant reference number | Author |
| --- | --- | --- |
| Howard Hughes Medical Institute | | Alejandro Sánchez Alvarado |
| National Science Foundation | 1923372 | Nicolas Rohner |
| National Institutes of Health | GM127872 | Nicolas Rohner |
| Stowers Institute for Medical Research | | Andrew C Box Christopher Wood Alejandro Sánchez Alvarado Nicolas Rohner |
| Deutsche Forschungsgemeinschaft | PE 2807/1-1 | Robert Peuß |

| American Association of Anatomists | Postdoctoral fellowship | Alice Accorsi |
| National Institutes of Health | DP2AG071466 | Nicolas Rohner |
| Society for Developmental Biology | Emerging Research Organisms Grant | Alice Accorsi |
| Edward Mallinckrodt Foundation | | Nicolas Rohner |

The funders had no role in study design, data collection and interpretation, or the decision to submit the work for publication.

### Author contributions

Alice Accorsi, Robert Peuß, Conceptualization, Data curation, Formal analysis, Supervision, Funding acquisition, Visualization, Methodology, Writing - original draft, Project administration, Writing - review and editing; Andrew C Box, Conceptualization, Data curation, Formal analysis, Investigation, Visualization, Methodology, Writing - review and editing; Christopher Wood, Software, Formal analysis, Methodology; Alejandro Sánchez Alvarado, Nicolas Rohner, Conceptualization, Supervision, Funding acquisition, Project administration, Writing - review and editing

### Author ORCIDs

Alice Accorsi (iD) https://orcid.org/0000-0003-0606-2977
Robert Peuß (iD) https://orcid.org/0000-0002-9716-6650
Alejandro Sánchez Alvarado (iD) https://orcid.org/0000-0002-1966-6959
Nicolas Rohner (iD) https://orcid.org/0000-0003-3248-2772

### Ethics

Animal experimentation: Research and animal care were approved by the Institutional Animal Care and Use Committee (IACUC) of the Stowers Institute for Medical Research. protocol (#2018-0187 and #2019-080).

### Decision letter and Author response

Decision letter https://doi.org/10.7554/eLife.65372.sa1
Author response https://doi.org/10.7554/eLife.65372.sa2

## Additional files

### Supplementary files

• Supplementary file 1. Features used for the morphology assay. Names and descriptions of the features quantified by IDEAS software and used for clustering events based on cell morphology in the homeostasis cell composition experiment. BF: brightfield; CI: cell intrinsic; CF: cell function.

• Supplementary file 2. Features used for the phagocytosis assay. Names and descriptions of the features quantified by IDEAS software and used for clustering events based on cell morphology and function in the phagocytosis experiment. BF: brightfield; CI: cell intrinsic; CF: cell function.

• Supplementary file 3. Cell cluster properties. Cell cluster properties (*e.g.*, cell numbers per cluster, cell features used for clustering, feature values for each cluster) for zebrafish whole kidney marrow (WKM) morphology and phagocytosis assay and for *P. canaliculata* hemocyte morphology and phagocytosis assay.

• Supplementary file 4. Cell gallery for zebrafish whole kidney marrow (WKM) in homeostasis condition. Representative cell images belonging to each individual cluster identified by Image3C for zebrafish WKM in homeostasis condition are shown. BF: brightfield; SSC: side scatter signal; Draq5: nuclear staining. Merge represents the overlay of BF, SSC, and Draq5.

• Supplementary file 5. Phagocytosis vs. phagocytosis inhibited with CCB on zebrafish whole kidney marrow (WKM). Results of negative binomial regression analysis comparing cluster relative

abundance between phagocytosis samples (CTV-*S. aureus*) and phagocytosis inhibited with CCB samples (CTV-*S. aureus* + CCB) in the zebrafish phagocytosis experiment. FC: fold change; CPM: count per million; LR: likelihood ratio; FDR: false discovery rate. Relative graph is reported in *Figure 3B*.

• Supplementary file 6. Phagocytosis vs. phagocytosis inhibited with ice on zebrafish whole kidney marrow (WKM). Results of negative binomial regression analysis comparing cluster relative abundance between phagocytosis samples (CTV-*S. aureus*) and phagocytosis inhibited with ice samples (CTV-*S. aureus* + Ice) in the zebrafish phagocytosis experiment. FC: fold change; CPM: count per million; LR: likelihood ratio; FDR: false discovery rate. Relative graph is reported in *Figure 3—figure supplement 2*.

• Supplementary file 7. Cell gallery for zebrafish whole kidney marrow (WKM) after phagocytosis assay. Representative cell images belonging to each individual cluster identified by Image3C for zebrafish WKM after phagocytosis assay are shown. BF: brightfield; DHR: fluorescent reactive oxygen species indicator; SSC: side scatter signal; Bac: CTV signal (*S. aureus* labeling); Draq5: nuclear staining. Merge represents the overlay of DHR, Bac, and Draq5.

• Supplementary file 8. Cell gallery for *P. canaliculata* hemocytes in homeostasis condition. Representative cell images belonging to each individual cluster identified by Image3C for snail hemocytes in homeostasis condition are shown. Ch01 is brightfield, Ch06 is side scatter signal, and Ch11 is Draq5 (nuclear staining). Merge represents the overlay of Ch01, Ch06, and Ch11.

• Supplementary file 9. Phagocytosis vs. phagocytosis inhibited with EDTA on *P. canaliculata* hemocytes. Results of negative binomial regression analysis comparing cluster relative abundance between phagocytosis samples (CTV-*S. aureus*) and phagocytosis inhibited with EDTA samples (CTV-*S. aureus* + EDTA) in the apple snail *P. canaliculata* phagocytosis experiment. FC: fold change; CPM: count per million; LR: likelihood ratio; FDR: false discovery rate. Relative graph is reported in *Figure 5B*.

• Supplementary file 10. Phagocytosis vs. phagocytosis inhibited with ice on *P. canaliculata* hemocytes. Results of negative binomial regression analysis comparing cluster relative abundance between phagocytosis samples (CTV-*S. aureus*) and phagocytosis inhibited with ice samples (CTV-*S. aureus* + Ice) in the apple snail *P. canaliculata* phagocytosis experiment. FC: fold change; CPM: count per million; LR: likelihood ratio; FDR: false discovery rate. Relative graph is reported in *Figure 5—figure supplement 2*.

• Supplementary file 11. Cell gallery for *P. canaliculata* hemocytes after phagocytosis assay. Representative cell images belonging to each individual cluster identified by Image3C for snail hemocytes after phagocytosis assay are shown. Ch01 is brightfield, Ch02 is DHR signal (reactive oxygen species indicator), Ch06 is side scatter signal, Ch07 is CTV signal (*S. aureus* labeling), and Ch11 is Draq5 (nuclear staining). Merge represents the overlay of Ch02, Ch06, Ch07, and Ch11.

• Transparent reporting form

### Data availability

All original data underlying this manuscript can be accessed from the Stowers Original Data Repository at http://www.stowers.org/research/publications/libpb-1390. Image3C code and description are freely available at the GitHub repository https://github.com/stowersinstitute/LIBPB-1390-Image3C.

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
