## [Decision Letter]

Thank you for submitting your article "Image3C: a multimodal image-based and label independent integrative method for single-cell analysis" for consideration by *eLife*…. Your article has been reviewed by 2 peer reviewers, one of whom is a member of our Board of Reviewing Editors, and the evaluation has been overseen by Didier Stainier as the Senior Editor.

Both reviewers have agreed that your submitted manuscript is a valuable contribution to the field of label-free assessment of complex cellular mixtures and should be published given some crucial revisions.

Essential revisions:

1) Compare and contrast Image3C with other label-free phenotyping approaches, an issue raised by all reviewers.

2) The software should be made more accessible for the general user.

3) Please address the concerns regarding the assignment post CNN being "unsupervised", which was raised by both reviewers.

4) Address other options for Image3C workflow tasks (e.g. feature extraction, image processing, clustering, etc.). – At a minimum, explain the choices more critically

Also, please address further comments below and correct/clarify figures as appropriate (see comments)

*Reviewer #1 (Recommendations for the authors):*

– Given this is a tool and resource paper, I think more attention should be placed in making the software accessible to the general users. In the github page, there are multiple mentions of 'If on site at SIMR' (which I assume refer to the Stower Institute for Medical Research), which did not inspire confidence that this tool is designed for use by a wide audience.

– Can the authors compare and clarify the contribution of Image3C against other label-free single cell image-based phenotyping systems?

– Since there are many choice of algorithms in each step of the pipeline, such as feature extraction, image processing, clustering, etc. Have the authors performed a comparison of several sensible methods to show that their choice is the most suitable? At the minimum, the authors should explain their choice more critically.

*Reviewer #2 (Recommendations for the authors):*

The authors present a convincing argument for the use of imaging flow cytometer data in profiling and comparing complex cell mixtures, such as dissociated tissues. I especially appreciated the application of Image3C to separate datasets from fish (*D. rerio* WKM) and a non-model system (Hemolymph from the apple snail): Not only do they present consistent clustering, but also the emergence of phagocytes upon infection. I can appreciate that Image3C is not only applicable without a host of established reagents, but that it might add an additional layer based on cell morphology, rather than pure transcriptomics (or genome accessibility) – these layers of information may be complimentary, which I suggest the authors make a point of.

The code is largely available (except the steps using proprietary software), which is good. However, I believe the code is currently too disjointed to be useful to a non-expert. The workflow currently involves steps using the Amnis IDEAS package, followed by custom scripts in R, followed by VorteX (I believe in Java) followed by more scripts in R, followed by optional classification using CNNs in python, with a side branch of operations using commercial flow cytometry software (FCSexpress). I think it would be advisable to somehow package this assemblage of scripts into a more accessible and user-friendly package. For example, open source packages like FlowCore might be integratable.

Along the same lines, are there any alternatives for depending on the specific instrument (Amnis ImageStream Mark2) and the proprietary IDEAS software?

As is, I was unable to test the packages for lack of instrument-specific data, software.

Sample data should be provided together with a testable, streamlined software package as well.

The authors have not sufficiently contextualized published work on label-free extraction of informative image features. Nor have they compared Image3C performance !

The authors should highlight and illustrate the reproducibility of the data across independent data sets (assuming this is true) – After going through the Materials and methods, I realized that several replicates were generated for each data set. Supplemental figures attest to that.

I am still not clear on how exactly cell type identities were assigned to the FDL clusters. It is my understanding that a lot of prior knowledge (and corresponding studies) was (were) necessary for assignment. As such, analysis using Image3C may be less useful for assessing cell type complexity, but rather for changes therein. The claim of "cell types are identified by de novo clustering" should then be toned down.

Furthermore, the statement that

"…this produces a CNN-based cell classifier 'machine' used to quantify subsequently acquired image-based flow cytometry data and to compare cellular composition of samples across multiple experiments, in a high-throughput and unsupervised manner…„

is utterly bewildering to me. The CNN utilized cluster features as assignment guides. I believe that this is supervised by definition."

I was unable to sufficiently evaluate some of the figures, in particular the imaged cell arrays in various channels. In many cases the levels are such that signal I have to assume to be there is not visible. This is particularly true for the Phagocytosis assay, where DHR and CTV signal should be visible. Also, in these cases a direct comparison with sister cluster cells should be shown.

Check labels (especially in supplemental figures) for clarity (e.g. "Single and Nuc" in S3?)

---

## [Author Response]

Essential revisions:1) Compare and contrast Image3C with other label-free phenotyping approaches, an issue raised by all reviewers.

We included in the manuscript a new table (Table 1) where we compare label-free phenotyping and cell clustering approaches. We took into consideration on which samples the tool was tested, the need of prior knowledge of the sample and/or species-specific reagents at any point of the process, and the hardware and software required. We also further discussed these aspects in the main text.

2) The software should be made more accessible for the general user.

To make our tool more accessible, we have: (1) thoroughly revised the readme document in the GitHub page making it clearer and more detailed; (2) added tutorial videos to the GitHub page to show how to use some of the mentioned software; (3) saved the R code as a markdown page to make it more user friendly; (4) renamed the example files that can be used to test our pipeline to make them more self-explanatory; (5) saved the Python code in the more user friendly Jupyter Notebooks; and (6) transformed Figure 1—figure supplement 1 in an interactive map where it is possible to click on the different steps of the Image3C pipeline and be automatically directed to the corresponding section of our GitHub repository page.

3) Please address the concerns regarding the assignment post CNN being "unsupervised", which was raised by both reviewers.

We thank both reviewers for pointing this out and we apologize for the mistake. In the text, we rephrased all the sentences in which the CNN was mistakenly defined as “unsupervised”.

4) Address other options for Image3C workflow tasks (e.g. feature extraction, image processing, clustering, etc.). – At a minimum, explain the choices more critically.

We thank the reviewers for this comment. We did not perform a side-by-side comparison between several methods for each step of the pipeline, but we chose the algorithms and software based on published literature, pilots we performed, and availability of options based on the task. We now include in the Material and Methods a more detailed explanation the criteria used for our choices and we now mention an alternative to FCS Express that is the only software used that is not free or open source.

Also, please address further comments below and correct/clarify figures as appropriate (see comments).Reviewer #1 (Recommendations for the authors):– Given this is a tool and resource paper, I think more attention should be placed in making the software accessible to the general users. In the github page, there are multiple mentions of 'If on site at SIMR' (which I assume refer to the Stower Institute for Medical Research), which did not inspire confidence that this tool is designed for use by a wide audience.

We completely agree with this comment and we apologize for the oversights. We removed all the references to SIMR and we included a number of changes aimed at making the pipeline more accessible. We have: (1) thoroughly revised the readme document in the GitHub page making it clearer and more detailed; (2) added tutorial videos to the GitHub page to show how to use some of the key software mentioned in the manuscript; (3) saved the R code as a markdown page to make it more user friendly; (4) renamed the example files that can be used to test our pipeline to make them more self-explanatory; (5) saved the Python code in the more user friendly Jupyter Notebooks; (6) transformed Figure 1-supplement figure 1 in an interactive map where it is possible to click on the different steps of the Image3C pipeline and be automatically directed to the corresponding section of our GitHub repository page.

– Can the authors compare and clarify the contribution of Image3C against other label-free single cell image-based phenotyping systems?

We included in the manuscript a new table (Table 1) where we compare label-free phenotyping and cell-clustering approaches. We took into consideration on which samples the tool was tested, the need for prior knowledge of the sample and/or species-specific reagents at any point of the process, and the hardware and software required. We also further discuss these aspects in the main text.

– Since there are many choice of algorithms in each step of the pipeline, such as feature extraction, image processing, clustering, etc. Have the authors performed a comparison of several sensible methods to show that their choice is the most suitable? At the minimum, the authors should explain their choice more critically.

We did not perform a side-by-side comparison between several methods for each step of the pipeline, but we chose the algorithms and software based on published literature, pilots we performed and availability of options based on the task. We now include in the Material and Methods a more detailed explanation for our choices and we mention an alternative to FCS Express which is the only software used that is not free or open source.

Reviewer #2 (Recommendations for the authors):The authors present a convincing argument for the use of imaging flow cytometer data in profiling and comparing complex cell mixtures, such as dissociated tissues. I especially appreciated the application of Image3C to separate datasets from fish (*D. rerio* WKM) and a non-model system (Hemolymph from the apple snail): Not only do they present consistent clustering, but also the emergence of phagocytes upon infection. I can appreciate that Image3C is not only applicable without a host of established reagents, but that it might add an additional layer based on cell morphology, rather than pure transcriptomics (or genome accessibility) – these layers of information may be complimentary, which I suggest the authors make a point of.

We thank the reviewer for this comment. We highlighted this point in the Introduction and in the Discussion, broadening up the scenarios where Image3C can be a valuable resource.

The code is largely available (except the steps using proprietary software), which is good. However, I believe the code is currently too disjointed to be useful to a non-expert. The workflow currently involves steps using the Amnis IDEAS package, followed by custom scripts in R, followed by VorteX (I believe in Java) followed by more scripts in R, followed by optional classification using CNNs in python, with a side branch of operations using commercial flow cytometry software (FCSexpress). I think it would be advisable to somehow package this assemblage of scripts into a more accessible and user-friendly package. For example, open source packages like FlowCore might be integratable.

To make our tool more accessible, we have: (1) thoroughly revised the readme document in the GitHub page making it clearer and more detailed; (2) added tutorial videos to the GitHub page to show how to use some of the mentioned software; (3) saved the R code as a markdown page to make it more user friendly; (4) renamed the example files that can be used to test our pipeline to make them more self-explanatory; (5) saved the Python code in the more user friendly Jupyter Notebooks; and (6) transformed Figure 1-supplement figure 1 in an interactive map where it is possible to click on the different steps of the Image3C pipeline and be automatically directed to the corresponding section of our GitHub repository page. We now include in the Material and Methods an explanation for our choices of software/algorithms and we mention an alternative to FCS Express that is the only software used that is not free or open source.

Along the same lines, are there any alternatives for depending on the specific instrument (Amnis ImageStream Mark2) and the proprietary IDEAS software? As is, I was unable to test the packages for lack of instrument-specific data, software. Sample data should be provided together with a testable, streamlined software package as well.

We thank the reviewer for this comment. We uploaded in the GitHub an example dataset for users to test the pipeline without having to collect data themselves. We also uploaded files saved at intermediate steps of analysis so the users can test the steps following the IDEAS software. Once the users are ready to perform the experiments and they have an Imagestream available, then the IDEAS software can be freely downloaded.

The authors have not sufficiently contextualized published work on label-free extraction of informative image features. Nor have they compared Image3C performance!

We have now included in the manuscript a new table (Table 1) where we compare label-free phenotyping and cell clustering approaches. We took into consideration on which samples the tool was tested, the need for prior knowledge of the sample and/or species-specific reagents at any point of the process, and the hardware and software required. We also further discuss these aspects in the main text.

Concerning the comparison of performance, Image3C was developed because of the lack of published tools able to perform de novo clustering without the use of specific cell-type markers. We also included, in the Material and Methods, the rationale for selecting these specific algorithms as part of the pipeline and, in the Discussion, we now mention that the true positive rate we obtain after CNN training is comparable to the rate reported in other trained neural networks.

The authors should highlight and illustrate the reproducibility of the data across independent data sets (assuming this is true) – After going through the Materials and methods, I realized that several replicates were generated for each data set. Supplemental figures attest to that.

We thank the reviewer for mentioning the replicates aspects. The number of replicates per datasets are reported in the figure legends and in the Material and Methods. We now include comments about the reproducibility between replicates in the Results and we include 3 new supplementary figures, that, together with Figure 2—figure supplement 2, illustrate the correlation between samples and potential outliers (Figure 3—figure supplement 1, Figure 4—figure supplement 1, and Figure 5—figure supplement 1).

I am still not clear on how exactly cell type identities were assigned to the FDL clusters. It is my understanding that a lot of prior knowledge (and corresponding studies) was (were) necessary for assignment. As such, analysis using Image3C may be less useful for assessing cell type complexity, but rather for changes therein. The claim of "cell types are identified by de novo clustering" should then be toned down.

We thank the reviewer for pointing this out. We completely agree that assigning cell types is a long-term goal and that additional experiments (such as functional assays) and examination of published literature will be necessary for achieving that. In the text, we toned down our statements, highlighting the possibility of comparing cluster abundance while making it clear that only a combination of cell morphologies, cell functions and comparison with data published in closely related organisms will allow for definitive cell type identification.

Furthermore, the statement that"…this produces a CNN-based cell classifier 'machine' used to quantify subsequently acquired image-based flow cytometry data and to compare cellular composition of samples across multiple experiments, in a high-throughput and unsupervised manner…„is utterly bewildering to me. The CNN utilized cluster features as assignment guides. I believe that this is supervised by definition."

We thank the reviewer for pointing this out. We completely agree and have removed the word “unsupervised” from the mentioned sentence.

I was unable to sufficiently evaluate some of the figures, in particular the imaged cell arrays in various channels. In many cases the levels are such that signal I have to assume to be there is not visible. This is particularly true for the Phagocytosis assay, where DHR and CTV signal should be visible. Also, in these cases a direct comparison with sister cluster cells should be shown.

We thank the reviewer and we apologize for the difficulties encountered in evaluating the signal in the cell images. Unfortunately, FCS Express, the software that we used to combine cell images and their cluster IDs and produce the cell galleries (Supplementary Files 4, 7, 8 and 11) uses the information obtained from IDEAS and does not allow display adjustments (intensity/brightness/color) of the images. We agree that the color used for the bacterial signal is not giving the best contrast against the background and we apologize for the limited possibilities in adjusting the image features using FCS Express. As suggested from the reviewer, we included in the manuscript a new Supplementary Figure (Figure 5—figure supplement 4) showing in yellow the signal of the bacteria labelled with cell-trace-violett and comparing side-by-side professional phagocytes and non-phagocytic hemocytes from the apple snail. These images were obtained using the Gallery function of the Amnis IDEAS software and the value for the signal intensity is reported on top of each individual image. In addition, we now highlight in the legends of Figures 2, 3, 4 and 5 that a representation of cells belonging to each cluster has been shown in the Supplementary Files, where there is no space limitation.

Check labels (especially in supplemental figures) for clarity (e.g. "Single and Nuc" in S3 ?)

We checked all the labels in the Figures, Supplementary Files and we modified those that could be unclear to readers and users, such as labels in Figure 2—figure supplement 2 (previous Figure S3) and Supplementary File 3.